# Dynamic compartmentalization of the pro-invasive transcription factor NHR-67 reveals a role for Groucho in regulating a proliferative-invasive cellular switch in *C. elegans*

Taylor N Medwig-Kinney[1]*[†], Brian A Kinney[2][†], Michael AQ Martinez[1], Callista Yee[3], Sydney S Sirota[1][‡], Angelina A Mullarkey[1], Neha Somineni[1][§], Justin Hippler[1,4][#], Wan Zhang[1], Kang Shen[3], Christopher Hammell[2], Ariel M Pani[5], David Q Matus[1]*[¶]

[1]Department of Biochemistry and Cell Biology, Stony Brook University, Stony Brook, United States; [2]Cold Spring Harbor Laboratory, Cold Spring Harbor, United States; [3]Howard Hughes Medical Institute, Department of Biology, Stanford University, Stanford, United States; [4]Science and Technology Research Program, Smithtown High School East, St. James, United States; [5]Departments of Biology and Cell Biology, University of Virginia, Charlottesville, United States

*For correspondence:
tmkinney@unc.edu (TNM-K);
david.matus@stonybrook.edu
(DQM)

Present address: [†]Department of Biology, University of North Carolina at Chapel Hill, Chapel Hill, United States; [‡]Touro College of Osteopathic Medicine, Middletown, United States; [§]Integra LifeSciences, Princeton, United States; [#]Northeastern University, Boston, United States; [¶]Arcadia Science, Berkeley, United States

**Abstract** A growing body of evidence suggests that cell division and basement membrane invasion are mutually exclusive cellular behaviors. How cells switch between proliferative and invasive states is not well understood. Here, we investigated this dichotomy in vivo by examining two cell types in the developing *Caenorhabditis elegans* somatic gonad that derive from equipotent progenitors, but exhibit distinct cell behaviors: the post-mitotic, invasive anchor cell and the neighboring proliferative, non-invasive ventral uterine (VU) cells. We show that the fates of these cells post-specification are more plastic than previously appreciated and that levels of NHR-67 are important for discriminating between invasive and proliferative behavior. Transcription of NHR-67 is downregulated following post-translational degradation of its direct upstream regulator, HLH-2 (E/Daughterless) in VU cells. In the nuclei of VU cells, residual NHR-67 protein is compartmentalized into discrete punctae that are dynamic over the cell cycle and exhibit liquid-like properties. By screening for proteins that colocalize with NHR-67 punctae, we identified new regulators of uterine cell fate maintenance: homologs of the transcriptional co-repressor Groucho (UNC-37 and LSY-22), as well as the TCF/LEF homolog POP-1. We propose a model in which the association of NHR-67 with the Groucho/TCF complex suppresses the default invasive state in non-invasive cells, which complements transcriptional regulation to add robustness to the proliferative-invasive cellular switch in vivo.

## eLife assessment

This **valuable** data study presents **convincing** data that expression of the *C. elegans* transcription factor NHR-67 is sufficient to drive an invasive fate, and that the alternative proliferative fate is associated with NHR-67 transcriptional down-regulation. While the observation that NHR-67 forms punctae associated with transcriptional repressors in non-invasive cells is intriguing, the work does not yet established a clear link between the formation and dissolution of NHR-67 condensates with the activation of downstream genes that NHR-67 is actively repressing. The work will be of interest to developmental biologists studying transcriptional control of cell fate specification in animals,

especially once issues around the functional significance of the NHR-67 containing punctae are resolved.

## Introduction

Cellular proliferation and invasion are key aspects of development (reviewed in *Medwig and Matus, 2017*), and are also two of the defining hallmarks of cancer (reviewed in *Hanahan and Weinberg, 2000*). A growing body of evidence suggests that cell cycle progression and invasion through a basement membrane are mutually exclusive cellular behaviors in both development and disease states (reviewed in *Kohrman and Matus, 2017*). Switching between invasive and proliferative phenotypes has been observed in melanoma and recently in breast cancer (*Hoek et al., 2008*; *Mondal et al., 2022*), but how these cell states are regulated in the context of development is not well understood. To investigate how this dichotomy in cellular behavior is controlled in vivo, we used *C. elegans*, leveraging its highly stereotypical development (*Sulston and Horvitz, 1977*), as well as its genetic and optical tractability. During the development of the hermaphroditic reproductive system, the proximal granddaughters of the Z1 and Z4 somatic gonad progenitors, Z1.pp and Z4.aa, give rise to four cells that will adopt one of two cellular fates: a proliferative VU cell or the terminally differentiated, invasive anchor cell (AC) (*Figure 1A*; *Kimble and Hirsh, 1979*). The distal cells of this competency group, Z1.ppa and Z4.aap, quickly lose their bipotentiality and become VU cells (*Seydoux et al., 1990*). In contrast, the proximal cells, Z1.ppp and Z4.aaa, undergo a stochastic Notch-mediated cell fate decision, giving rise to another VU cell and the post-mitotic AC (*Figure 1A and B*; *Greenwald et al., 1983*; *Seydoux and Greenwald, 1989*). Following fate specification, the AC undergoes invasive differentiation and breaches the underlying basement membrane, connecting the uterus to the vulval epithelium to facilitate egg-laying (*Figure 1B*; *Sherwood and Sternberg, 2003*). The AC is the default cell fate of the somatic gonad, as disruption of Notch or Wnt signaling results in ectopic AC specification (*Phillips et al., 2007*; *Seydoux and Greenwald, 1989*). Therefore, AC fate must be actively repressed.

Our previous work has shown that AC invasion is dependent on $G_0$ cell cycle arrest, which is coordinated by the pro-invasive transcription factor NHR-67 (NR2E1/TLX) (*Figure 1—figure supplement 1A*; *Matus et al., 2015*). NHR-67 functions within a gene regulatory network comprised of four conserved transcription factors whose homologs have been implicated in several types of metastatic cancer (*Liang and Wang, 2020*; *Milde-Langosch, 2005*; *Nelson et al., 2021*; *Wang and Baker, 2015*). We previously reported that NHR-67 is regulated by a feed-forward loop formed by EGL-43 (Evi1) and HLH-2 (E/Daughterless), which functions largely in parallel to a cell cycle-independent subcircuit controlled by FOS-1 (Fos) (*Figure 1—figure supplement 1A*; *Medwig-Kinney et al., 2020*). EGL-43, HLH-2, and NHR-67 are reiteratively used within the Z lineage of the somatic gonad, in that, they also function to independently regulate LIN-12 (Notch) signaling during the initial AC/VU cell fate decision (*Medwig-Kinney et al., 2020*). Despite its role in lateral inhibition between Z1.ppp and Z4.aaa, expression of LIN-12 is not absolutely required for VU cell fate (*Sallee et al., 2015a*). Cell cycle state also cannot explain the difference between AC and VU cell fates, as arresting VU cells in $G_0$ through ectopic expression of CKI-1 (p21/p27) does not make them invasive (*Smith et al., 2022*). Thus, the mechanisms responsible for maintaining AC and VU cellular identities following initial cell fate specification remain unclear.

Maintenance of differentiated cell identity is essential for ensuring tissue integrity during development and homeostasis, and the inability to restrict phenotypic plasticity is now being recognized as an integral part of cancer pathogenesis (*Hanahan, 2022*). In vitro studies have identified several factors that safeguard differentiated cell identity (reviewed in *Brumbaugh et al., 2019*). Despite its largely autonomous modality of development, *C. elegans* has emerged as an ideal model system to study cell fate maintenance in vivo. There have been several reports of cell fate transformations that occur naturally, including two epithelial-to-neural transdifferentiation events (*Jarriault et al., 2008*; *Riva et al., 2022*), or following fate challenges (reviewed in *Rothman and Jarriault, 2019*). In such contexts, several epigenetic factors, including chromatin remodelers and histone chaperones, have been identified for their roles in restricting cell fate reprogramming (*Hajduskova et al., 2019*; *Kagias et al., 2012*; *Kolundzic et al., 2018*; *Patel et al., 2012*; *Rahe and Hobert, 2019*; *Zuryn et al., 2014*). However, in some cases, ectopic expression of a specific transcription factor is sufficient to overcome

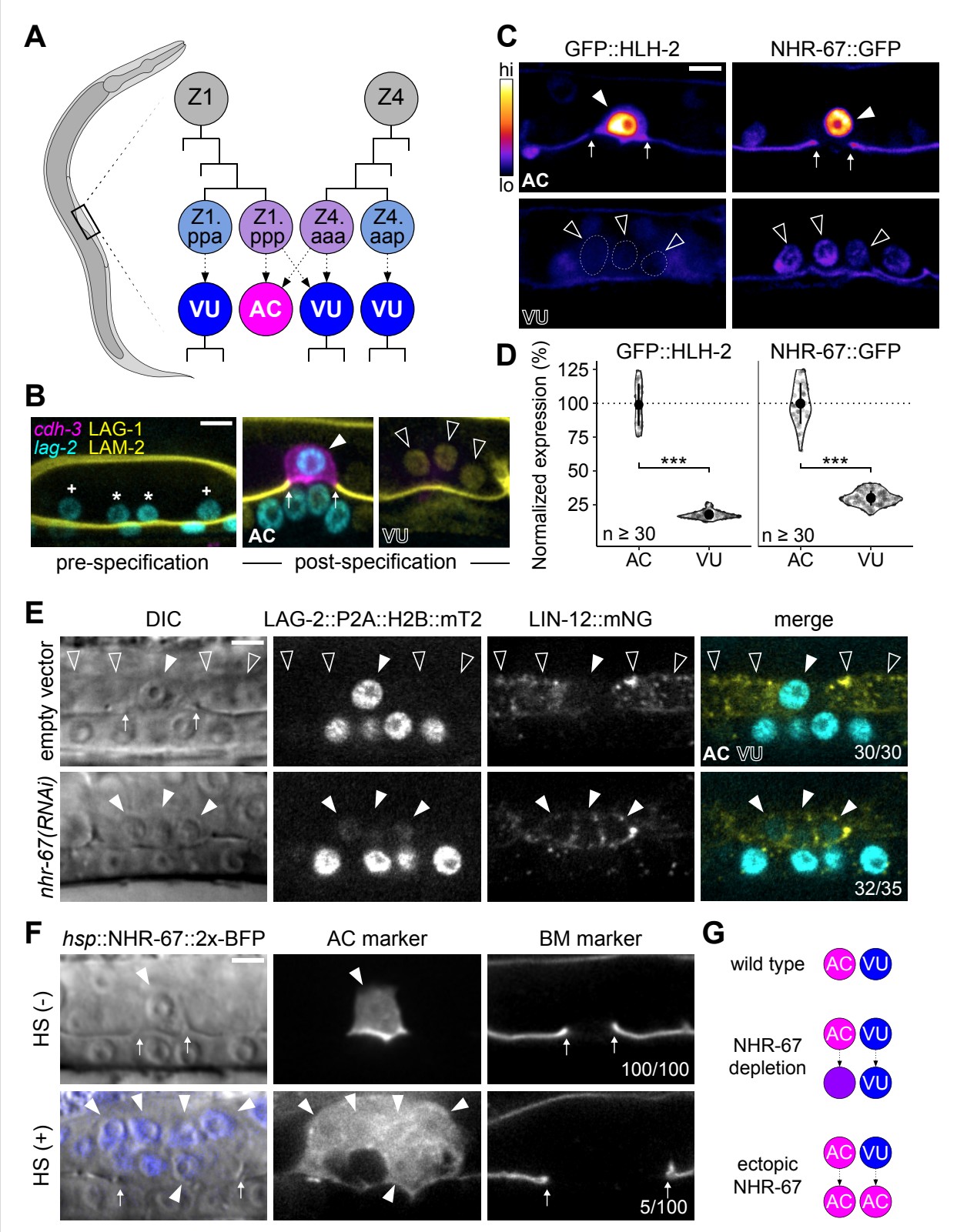

**Figure 1.** Invasive AC fate correlates to high levels of NHR-67. (**A**) Schematic of *C. elegans* anchor cell (AC, magenta) and ventral uterine (VU, blue) cell fate specification from the Z1 and Z4 somatic gonad precursor cell lineages (p, posterior daughter; a, anterior daughter). (**B**) Micrographs depicting AC and VU cell differentiation over developmental time. AC/VU precursors express LAG-2 (H2B::mTurquoise), which eventually becomes restricted to the AC, whereas VU cells express LAG-1 (mNeonGreen) post-specification. The differentiated AC (*cdh-3p*::mCherry::moeABD) then invades through

*Figure 1 continued on next page*

*Figure 1 continued*

the underlying basement membrane (LAM-2::mNeonGreen). (**C–D**) Representative heat map micrographs (**C**) and quantification (**D**) of GFP-tagged HLH-2 and NHR-67 expression in the AC and VU cells at the time of AC invasion. (**E**) Expression of Notch (*lin-12*::mNeonGreen) and Delta (*lag-2*::P2A::H2B::mTurquoise2) following RNAi-induced knockdown of NHR-67 compared to empty vector control. (**F**) Micrographs depicting the ectopic invasive ACs (*cdh-3p*::mCherry::moeABD, arrowheads) and expanded basement membrane (*laminin*::GFP, arrows) gap observed following heat shock-induced expression of NHR-67 (*hsp*::NHR-67::2x-BFP) compared to non-heat shocked controls. (**G**) Schematic summarizing AC and VU cell fates that result from perturbations of NHR-67 levels. For all figures: asterisk (*), AC/VU precursor; plus (+), VU precursor; solid arrowhead, AC; open arrowhead, VU cell; arrows, basement membrane breach. Statistical significance determined by Student's t-test (*$p > 0.05$, **$p > 0.01$, ***$p > 0.001$). Scale bars, 5 µm.

The online version of this article includes the following source data and figure supplement(s) for figure 1:

**Source data 1.** Raw data of GFP-tagged transcription factor expression in the anchor cell (AC) and ventral uterine (VU) cells, as reported in *Figure 1C and D* and *Figure 1—figure supplement 1B and C*.

**Source data 2.** Raw data of LAG-2::P2A::H2B::mTurquoise2 and LIN-12::mNeonGreen expression in NHR-67-deficient anchor cells (ACs) compared to control AC and ventral uterine (VU) cells, as reported in *Figure 1E* and *Figure 1—figure supplement 2A and B*.

**Figure supplement 1.** Expression of pro-invasive transcription factors EGL-43 and FOS-1 in the somatic gonad.

**Figure supplement 2.** NHR-67-deficient anchor cells (ACs) express both Notch and Delta.

these barriers, as was first shown through pioneering work in mouse embryonic fibroblasts (*Davis et al., 1987*). Indeed, there are several examples in *C. elegans* where ectopic expression of single lineage-specific transcription factors induces cell fate transformations (*Fukushige and Krause, 2005*; *Gilleard and McGhee, 2001*; *Horner et al., 1998*; *Jin et al., 1994*; *Kiefer et al., 2007*; *Quintin et al., 2001*; *Richard et al., 2011*; *Riddle et al., 2013*; *Tursun et al., 2011*; *Zhu et al., 1998*). Moreover, *C. elegans* uterine tissue may be particularly amenable to fate transformations, as ectopic expression of a single GATA transcription factor, ELT-7, is sufficient to induce transorganogenesis of the somatic gonad into the gut by reprogramming the mesodermally-derived tissue into endoderm (*Riddle et al., 2016*). Valuable insights have been made into how the function of fate-specifying transcription factors can be tuned through means such as autoregulation and dynamic heterodimerization (*Leyva-Díaz and Hobert, 2019*; *Sallee et al., 2017*). We are just beginning to understand how an additional layer of control over transcriptional regulators can be achieved through compartmentalization (*Boija et al., 2018*; *Lim and Levine, 2021*).

Here, in our endeavor to understand how AC and VU cellular fates are maintained, we identified two mechanisms that together modulate the invasive-proliferative switch in *C. elegans*. We found that high levels of NHR-67 expression are sufficient to drive invasive differentiation, and that NHR-67 is transcriptionally downregulated in the non-invasive VU cells following the post-translational degradation of its direct upstream regulator, HLH-2. Additionally, we observed that the remaining NHR-67 protein in the VU cells compartmentalizes into punctae that exhibit liquid-like properties including dynamic assembly, fusion, and dissolution over the cell cycle as well as rapid recovery kinetics after photobleaching. These NHR-67 punctae colocalize in vivo with UNC-37 and LSY-22, homologs of the transcriptional co-repressor Groucho, as well as with POP-1 (TCF/LEF), which is likely mediated through a direct interaction between UNC-37 and the intrinsically disordered C-terminal region of NHR-67. Through functional perturbations, we demonstrate that UNC-37, LSY-22, and POP-1 contribute to the repression of the default invasive state in VU cells. We propose a model in which NHR-67 compartmentalizes through its interaction with Groucho, which, combined with transcriptional downregulation of NHR-67, may suppress invasive differentiation.

## Results

### NHR-67 expression levels are important for distinguishing AC and VU cell identity

Despite arising from initially equipotent cells, the differentiated AC and VU cells exhibit very distinct cellular behaviors. The AC terminally differentiates to invade the underlying basement membrane while the VU cells remain proliferative, undergoing several rounds of division before terminally differentiating. One potential explanation for this difference in cell behavior is asymmetric expression of pro-invasive transcription factors. To investigate this possibility, we examined endogenous expression levels of four transcription factors that function in the gene regulatory network coordinating

AC invasion (EGL-43, FOS-1, HLH-2, and NHR-67) using GFP-tagged alleles (**Medwig-Kinney et al., 2020**). While FOS-1 levels are enriched in the AC compared to the VU cells (**Figure 1—figure supplement 1B and C**), FOS-1 has no known role in cell cycle regulation, so we did not pursue this protein further (**Medwig-Kinney et al., 2021**). EGL-43 was also not a promising candidate, as it is expressed in both cell types at comparable levels, with VU cells exhibiting approximately 89% of AC expression (**Figure 1—figure supplement 1B and C**). In contrast, HLH-2 exhibits significant asymmetry in expression, as VU cells express merely 17% of HLH-2 levels observed in the AC (**Figure 1C and D**). Previous studies have shown that dimerization-driven degradation of HLH-2 is responsible for its downregulation in the VU cells (**Benavidez et al., 2022**; **Karp and Greenwald, 2003**; **Sallee and Greenwald, 2015b**). NHR-67 exhibits a similar pattern of expression with over threefold enrichment in the AC, consistent with prior observations of transgenic reporters (**Figure 1C and D**; **Verghese et al., 2011**). Given the known role of NHR-67 in regulating cell cycle and invasion, we hypothesized that its differential expression between the AC and VU cells could contribute to their distinct cellular behaviors.

To assess the potential role of NHR-67 in regulating uterine cell identities, we manipulated its expression levels. We found that strong depletion of NHR-67 through RNA interference (RNAi) treatment results in ACs adopting VU-like characteristics. During AC/VU cell fate specification, LIN-12 (Notch) normally becomes restricted to the VU cells while the Delta-like ligand LAG-2 (visualized by LAG-2::P2A::H2B::mTurquoise2 **Medwig-Kinney et al., 2022**) accumulates in the AC (**Wilkinson et al., 1994**). Here, we observe that NHR-67 deficient ACs not only proliferated and failed to invade, as reported previously (**Matus et al., 2015**), but also ectopically expressed membrane-localized Notch (visualized by LIN-12::mNeonGreen **Pani et al., 2022**; **Figure 1E**; **Figure 1—figure supplement 2A and B**). Notably, NHR-67-deficient ACs expressed both LIN-12 and LAG-2, potentially indicating an intermediate state between AC and VU cell fate (**Figure 1E**; **Figure 1—figure supplement 2A and B**). Next, we ectopically expressed NHR-67 ubiquitously following AC/VU specification using a heat shock inducible transgene (*hsp*::NHR-67::2x-BFP) (**Medwig-Kinney et al., 2020**) and observed the presence of multiple invasive ACs at a low penetrance (approximately 5%, n>50), denoted by ectopic expression of an AC marker (*cdh-3p*::mCherry::moeABD) and expansion of the basement membrane gap (**Figure 1F**). As previous work has demonstrated that proliferative ACs cannot invade (**Matus et al., 2015**), we concluded that these invasive ectopic ACs most likely arose from the fate conversion of neighboring VU cells. Taken together, these pieces of evidence suggest that high and low levels of NHR-67 correlate to properties of AC and VU cell identities, respectively (**Figure 1G**).

## NHR-67 is enriched in the AC through transcriptional regulation by HLH-2

Next, we investigated how NHR-67 expression levels become asymmetric between the AC and VU cells. We and others have previously shown that HLH-2 positively regulates NHR-67 expression in the context of the AC (**Figure 1—figure supplement 1A**; **Bodofsky et al., 2018**; **Medwig-Kinney et al., 2020**). If this regulatory interaction also exists in the context of the VU cells, it could explain why the relative expression pattern of NHR-67 in the AC and VU cells mirrors that of HLH-2. In support of this hypothesis, we found that the initial onset of HLH-2, which has been shown to be asymmetric in Z1.pp and Z4.aa (**Attner et al., 2019**), correlates to onset of an NHR-67 transgene (**Figure 2—figure supplement 1A**; **Gerstein et al., 2010**). To test whether HLH-2 degradation is responsible for NHR-67 downregulation in the VU, we drove ectopic expression of HLH-2 using a transgene under the control of a heat shock inducible promoter (*hsp*::HLH-2::2x-BFP) (**Medwig-Kinney et al., 2020**) and observed elevated NHR-67 expression in VU cells (43% increase; n>30) (**Figure 2A and B**). To control against potential dimerization-driven degradation of HLH-2 in the VU cells, which the heat shock inducible transgene would still be susceptible to, we disrupted UBA-1, an E1 ubiquitin-activating enzyme that has recently been shown to be necessary for HLH-2 degradation in VU cells (**Benavidez et al., 2022**). Following perturbation of UBA-1 through RNAi treatment, HLH-2 expression in the VU cells increased more than fourfold and NHR-67 expression increased by nearly 60% compared to the empty vector control (**Figure 2—figure supplement 1B–D**). Both experiments suggest that NHR-67 expression in the VU cells is at least partially regulated by levels of HLH-2.

It has previously been proposed that the interaction between HLH-2 and NHR-67 is direct. This is based on the identification of E binding motifs within a 276 bp region of the NHR-67 promoter that is required for NHR-67 expression in the uterine tissue and encompasses the location of several

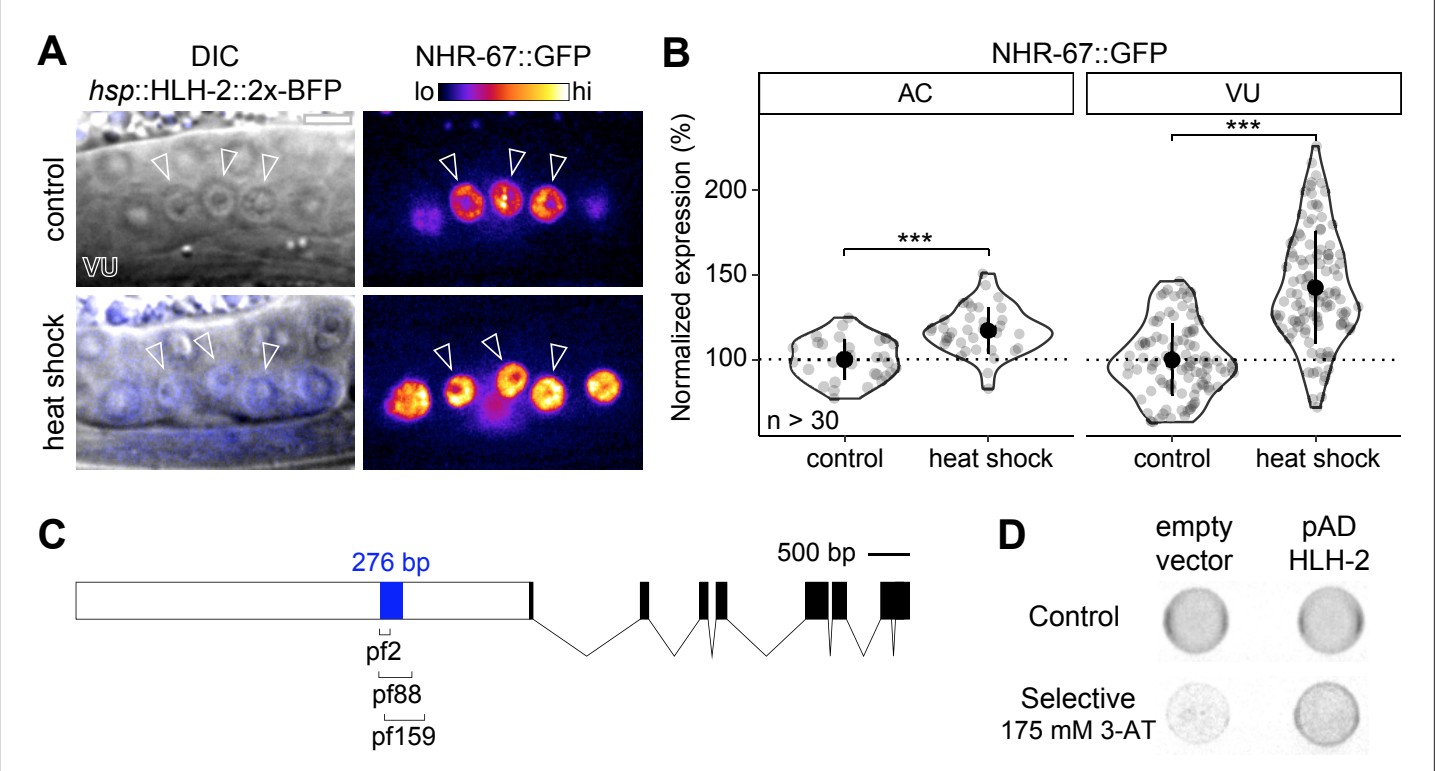

**Figure 2.** NHR-67 expression is downregulated in ventral uterine (VU) cells through direct transcriptional regulation by HLH-2. (A–B) Representative heat map micrographs (A) and quantification (B) of NHR-67::GFP expression in VU cells following heat shock-induced expression of HLH-2 (2x-BFP) compared to non-heat shocked controls. (C) Schematic of a 276 bp putative regulatory element within the promoter of NHR-67 (*Bodofsky et al., 2018*), annotated with the location of three hypomorphic mutations (*pf2*, pf88, and *pf159*). (D) Yeast one-hybrid experiment pairing HLH-2 Gal4-AD prey with the 276 bp fragment of the NHR-67 promoter as bait on SC-HIS-TRP plates with and without competitive inhibitor 3-AT (175 mM).

The online version of this article includes the following source data and figure supplement(s) for figure 2:

**Source data 1.** Raw data of NHR-67::GFP expression in the anchor cell (AC) and ventral uterine (VU) cells following heat-shock inducible expression of HLH-2, as reported in *Figure 2A and B*.

**Source data 2.** Raw data of GFP::HLH-2 and NHR-67::TagRFP-T expression in the anchor cell (AC) and ventral uterine (VU) cells following *uba-1* RNAi treatment compared to empty vector controls, as reported in *Figure 2—figure supplement 1B–D*.

**Figure supplement 1.** Onset of expression and regulatory interaction between NHR-67 and HLH-2 in the somatic gonad.

hypomorphic mutations (*pf2*, *pf88*, *pf159*) (*Figure 2C*; *Bodofsky et al., 2018*; *Verghese et al., 2011*). We performed a yeast one-hybrid assay by generating a bait strain containing this NHR-67 promoter region and pairing it with an HLH-2 Gal4-AD prey plasmid from an existing yeast one-hybrid library (*Reece-Hoyes et al., 2005*). Yeast growth on the selective SC-HIS-TRP plates containing the competitive inhibitor 3-aminotriazole (3-AT) suggests that HLH-2 is capable of binding directly to this 276 bp region of the NHR-67 promoter (*Figure 2D*). Together, these results support that direct transcriptional regulation of NHR-67 by HLH-2 contributes to the asymmetry in NHR-67 expression between the AC and VU cells.

## NHR-67 dynamically compartmentalizes in VU cell nuclei

Upon closer examination of GFP-tagged NHR-67, it became evident that the AC and VU cells not only exhibit differences in overall NHR-67 levels, but also in localization of the protein. While NHR-67 localization is fairly uniform throughout the AC nucleus (excluding the nucleolus), we often observed discrete punctae throughout the nuclei of VU cells (*Figure 3A and B*). These punctae were observed with NHR-67 endogenously tagged with several different fluorescent proteins, including GFP, mNeonGreen, mScarlet-I, and TagRFP-T (*Figure 3—figure supplement 1A and B*) and in the absence of tissue fixation methods that can cause artificial puncta (*Irgen-Gioro et al., 2022*), suggesting that they are not an artifact of the fluorophore or sample preparation.

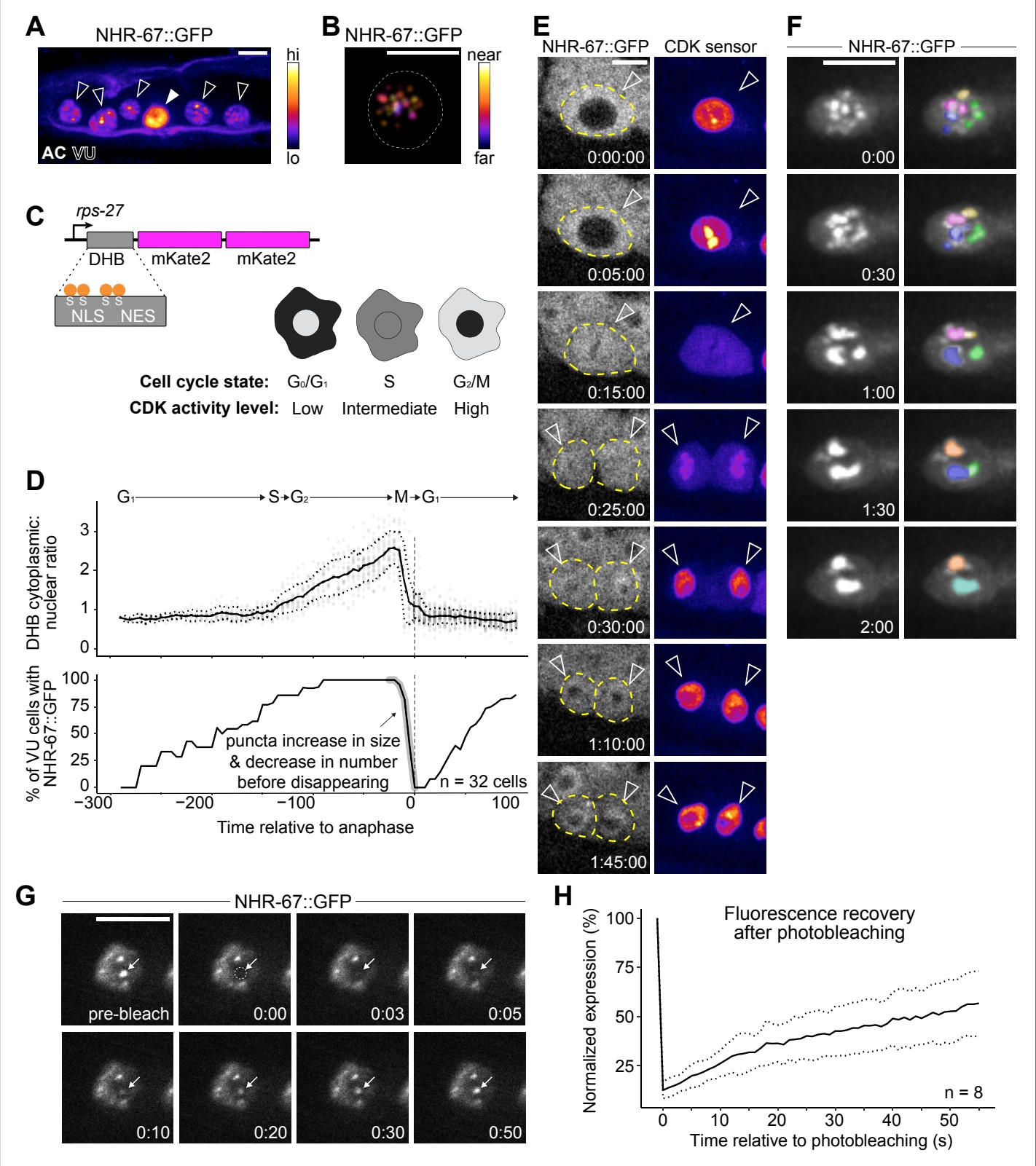

**Figure 3.** NHR-67 dynamically compartmentalizes in nuclei of ventral uterine (VU) cells. (**A**) Heat-map maximum intensity projection of NHR-67::GFP showing protein localization in the anchor cell (AC) and VU cells. (**B**) Spatial color-coded projection of NHR-67::GFP punctae in VU cells, with nuclear border indicated with a dotted line. (**C**) Schematic of DNA Helicase B (DHB) based CDK sensor and its dynamic localization over the cell cycle. (**D**) Graphs depicting CDK activity levels and corresponding cell cycle state (top), and percentage of cells exhibiting NHR-67::GFP punctae (bottom)

*Figure 3 continued on next page*

*Figure 3 continued*

over time, aligned to anaphase. (**E**) Representative time-lapse of NHR-67::GFP over the course of a cell cycle, with cell membranes indicated with dotted lines. (**F**) Time-lapse depicting NHR-67::GFP punctae fusion prior to cell division. Right panels are pseudo-colored. (**G–H**) Representative images (**G**) and quantification (**H**) depicting fluorescence recovery of NHR-67::GFP following photobleaching of individual punctae (arrow).

The online version of this article includes the following source data and figure supplement(s) for figure 3:

**Source data 1.** Raw data of CDK sensor (DHB) ratios in ventral uterine (VU) cells over time, as reported in *Figure 3D and E*.

**Source data 2.** Raw data of NHR-67::GFP puncta expression following photobleaching overtime, as reported in *Figure 3G and H*.

**Figure supplement 1.** Knock-in alleles of NHR-67.

To characterize the dynamics of these punctae during interphase states of the cell cycle, we paired GFP-tagged NHR-67 with a CDK activity sensor. The CDK activity sensor is comprised of a fragment of DNA Helicase B (DHB) fused to a fluorophore (2x-mKate2), expressed under a ubiquitous promoter (*Figure 3C*; *Adikes et al., 2020*). DHB contains a strong nuclear localization signal (NLS), flanked by four serine sites, as well as a weaker nuclear export signal (NES). As CDK activity increases over the cell cycle, the CDK sensor is translocated from the nucleus to the cytoplasm, allowing for correlation of its relative subcellular localization to the cell cycle state (*Figure 3C*; *Adikes et al., 2020*; *Spencer et al., 2013*). Time-lapse microscopy revealed that the number of NHR-67 punctae was dynamic over the course of the cell cycle, with punctae first appearing shortly after mitotic exit in the G1 phase, and then reducing in number to two large punctae prior to nuclear envelope breakdown before disappearing (*Figure 3D and E*). We collected additional recordings with finer time resolution and captured fusion, or condensation, of punctae prior to their dissolution (representative of 6 biological replicates) (*Figure 3F*). These punctae also exhibit relatively rapid diffusion kinetics, as observed by fluorescence recovery following photobleaching ($t_{1/2}$=46 s; n=8) at a rate within the same order of magnitude as P granule proteins PGL-1 and PGL-3 (*Figure 3G and H*; *Putnam et al., 2019*).

## Groucho homologs UNC-37 and LSY-22 associate with NHR-67 punctae and contribute to VU cell fate

Next, we tested the extent to which NHR-67 punctae colocalize with homologs of other proteins known to compartmentalize in nuclei by pairing GFP- and mScarlet-I-tagged NHR-67 with other endogenously tagged alleles. As NHR-67 is a transcription factor, we speculated that its punctae may represent clustering around sites of active transcription, which would be consistent with data showing RNA Polymerase II and the Mediator complex can compartmentalize with transcription factors (*Cho et al., 2018*). To test this hypothesis, we co-visualized NHR-67 with a GFP-tagged allele of *ama-1*, the amanitin-binding subunit of RNA polymerase II (*Hills-Muckey et al., 2022*) and failed to observe significant colocalization between NHR-67 and AMA-1 punctae (Manders' overlap coefficient, M=0.066) compared to negative controls where a single channel was compared to its 90-degree rotation (M=0.108) (*Figure 4A and B*). Another possibility considered is that NHR-67 punctae could correlate to chromatin organization, as heterochromatin has been shown to be compartmentalized in the nucleus (*Larson et al., 2017*; *Strom et al., 2017*). However, we did not observe significant colocalization of NHR-67 with the endogenously tagged HP1 heterochromatin proteins (*Patel and Hobert, 2017*) HPL-1 (M=0.076) or HPL-2 (M=0.083) (*Figure 4A and B*). We next tested if NHR-67 colocalizes with the transcriptional co-repressor Groucho, as Groucho had recently been shown to compartmentalize in the nuclei of cells in *Ciona* (*Treen et al., 2021*). The *C. elegans* genome encodes one Groucho homolog, UNC-37, which we acquired an mNeonGreen-tagged allele of *Ma et al., 2021*, and a Groucho-like protein, LSY-22, which we tagged with TagRFP-T (*Figure 4—figure supplement 1*). We observed significant colocalization of NHR-67 punctae with both LSY-22 (M=0.686) and UNC-37 (M=0.741), comparable to colocalization measures in heterozygous NHR-67::mScarlet-I/NHR-67::GFP animals (M=0.651), which were used as positive controls (*Figure 4A and B*). This evidence suggests that NHR-67 punctae do not localize to sites of active transcription or chromatin compaction, but instead associate with transcriptional co-repressors.

Since the AC is the default state of the AC/VU cell fate decision (*Seydoux and Greenwald, 1989*), we hypothesized that the punctae including NHR-67, UNC-37, and LSY-22 may function in repressing invasive differentiation. To test this hypothesis, we depleted UNC-37 and LSY-22 utilizing the auxin-inducible degron (AID) protein degradation system, in which a protein of interest is tagged with an

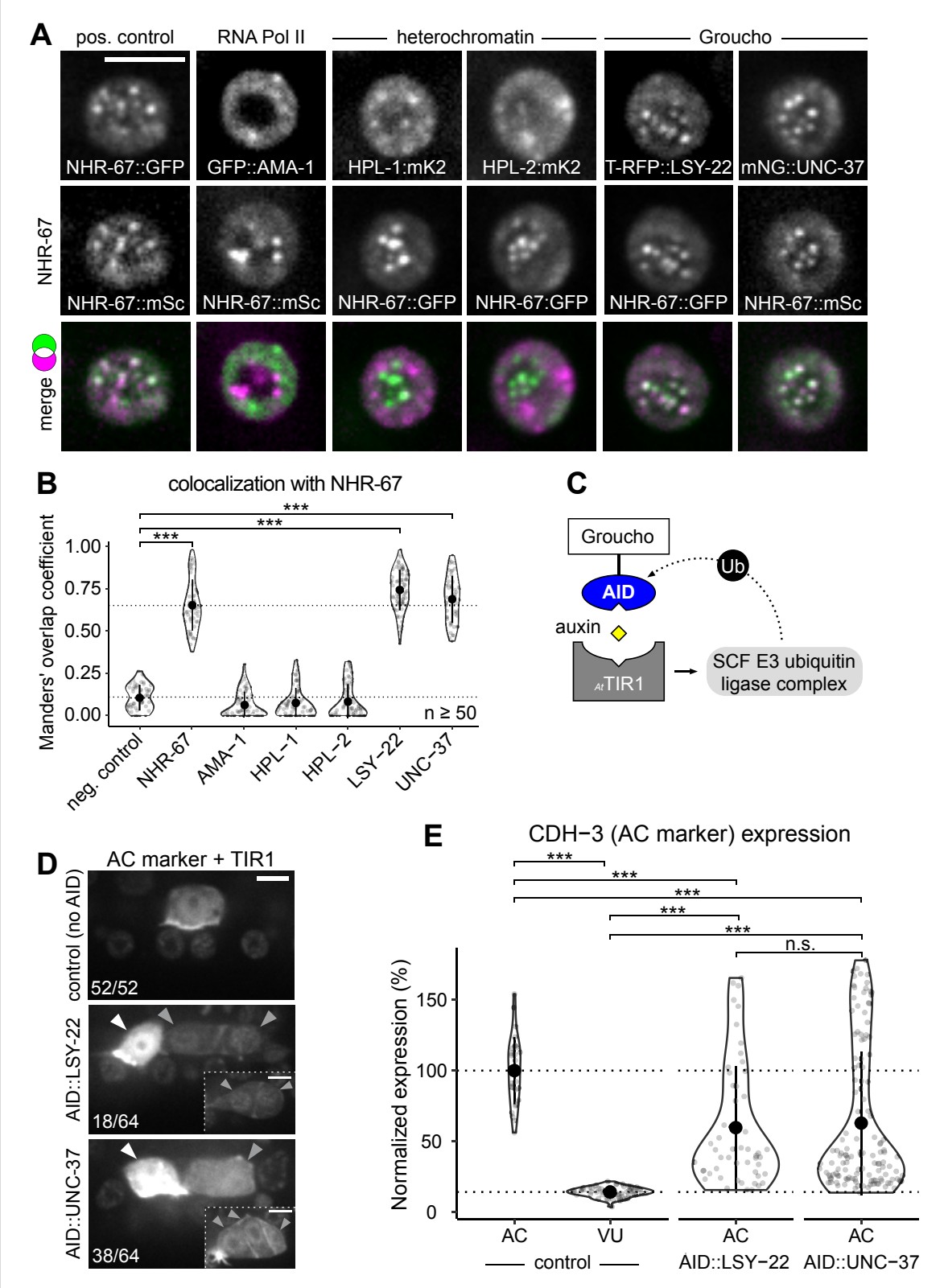

**Figure 4.** Groucho homologs LSY-22 an UNC-37 colocalize with NHR-67 punctae and contribute to maintenance of ventral uterine (VU) cell fate. (**A**) Co-visualization of NHR-67 with RNA Polymerase II (GFP::AMA-1), HP1 heterochromatin proteins (HPL-1::mKate2 and HPL-2::mKate2), and Groucho homologs (TagRFP-T::LSY-22 and mNeonGreen::UNC-37) in VU cells using endogenously tagged alleles. (**B**) Quantification of colocalization, with plot reporting Manders' overlap coefficients compared to negative controls (90-degree rotation of one channel) and positive controls. (**C**) Schematic

*Figure 4 continued on next page*

*Figure 4 continued*

of the auxin-inducible degron (AID) system, where $_{At}$TIR1 mediates proteasomal degradation of AID-tagged proteins in the presence of auxin. (**D**) Representative images of phenotypes observed following individual AID-depletion of UNC-37 and LSY-22 compared to control animals without AID-tagged alleles. All animals compared here are expressing TIR1 ubiquitously (*rpl-28p::*$_{At}$TIR1::T2A::mCherry::HIS-11) and an anchor cell (AC) marker (*cdh-3p*::mCherry::moeABD). Insets depict different z-planes of the same image. (**E**) Quantification of AC marker (*cdh-3p*::mCherry::moeABD) expression in ectopic ACs resulting from AID-depletion of UNC-37 and LSY-22 compared to control AC and VU cells.

The online version of this article includes the following source data and figure supplement(s) for figure 4:

**Source data 1.** Raw data of protein colocalization in ventral uterine (VU) cells, as reported in *Figure 4A and B* and *Figure 5D and E*.

**Source data 2.** Raw data of CDH-3 expression in the ectopic anchor cells (ACs) resulting from auxin-mediated depletion of AID-tagged LSY-22 or UNC-37, as reported in *Figure 4D and E*.

**Figure supplement 1.** Knock-in alleles of LSY-22.

**Figure supplement 2.** UNC-37 mutants show ectopic expression of anchor cell (AC) markers.

AID that is recognized by TIR1 in the presence of auxin and ubiquitinated by the SCF E3 ubiquitin ligase complex (*Figure 4C*; *Martinez et al., 2020*; *Zhang et al., 2015*). We re-tagged LSY-22 with mNeonGreen::AID (*Figure 4—figure supplement 1*) and acquired a BFP::AID-tagged allele of *unc-37* (*Kurashina et al., 2021*). Each AID-tagged allele was paired with a transgene encoding *Arabidopsis thaliana* TIR1 ($_{At}$TIR1) that was co-expressed with a nuclear-localized mCherry::HIS-11. Following auxin treatment, we observed ectopic expression of an AC marker (*cdh-3p*::mCherry::moeABD) in 28% of LSY-22::AID animals and 59% of UNC-37::AID animals (n=64 for both) (*Figure 4D and E*). These results are consistent with phenotypes we observed in genetic backgrounds with *unc-37* hypomorphic (*unc-37(e262wd26)*) and null (*unc-37(wd17wd22)*) mutant alleles (*Figure 4—figure supplement 2*). It is likely that dual depletion of UNC-37 and LSY-22 would result in a higher penetrance of ectopic ACs given their partial redundancy in function (*Flowers et al., 2010*), but animals possessing both AID-tagged alleles were not viable when paired with the $_{At}$TIR1 transgene.

## TCF/LEF homolog POP-1 associates with NHR-67 punctae and contributes to VU cell fate post-specification

While UNC-37 and LSY-22 appear to be important for the maintenance of normal uterine cell fates, both genes are broadly expressed and exhibit comparable levels (<10% difference) between the AC and VU cells (*Figure 5A and C*; *Figure 5—figure supplement 1A and B*); therefore, we hypothesized that another factor must be involved. It had previously been reported that the sole TCF/LEF homolog in *C. elegans*, POP-1, forms a repressive complex with UNC-37 in the early embryo to restrict expression of the endoderm-determining gene, END-1 (*Calvo et al., 2001*). Additionally, POP-1 has a known role in the development of the somatic gonad, as perturbing its function results in ectopic ACs (*Siegfried and Kimble, 2002*). Examination of an eGFP-tagged *pop-1* allele (*van der Horst et al., 2019*), showed significant enrichment in the VU cells (>25%) compared to the AC (*Figure 5B and C*; *Figure 5—figure supplement 1A and B*). We also observed that endogenous POP-1 forms punctae in the nuclei of VU cells, which had previously been observed during interphase in non-Wnt signaled embryonic cells (*Maduro et al., 2002*). These POP-1 punctae colocalize with NHR-67 (M=0.547), although to a lesser degree than UNC-37 and LSY-22, likely because the strong POP-1 fluorescence outside of punctae made them more difficult to segment (*Figure 5D and E*). Additionally, NHR-67(*RNAi*) treatment resulted in a significant increase in AC expression of eGFP::POP-1 compared to empty vector controls (225%, n>30), a pattern we observed following depletion of other transcription factors (*Medwig-Kinney et al., 2020*) and chromatin modifiers (*Smith et al., 2022*) required for AC arrest and invasion (*Figure 5F and G*; *Figure 5—figure supplement 2A and B*). This negative regulation of POP-1 by NHR-67 may explain why the proteins have opposite patterns of enrichment.

It has previously been suggested that POP-1 may be functioning as an activator in the VU precursors Z1.ppa and Z4.aap based on the relative expression of a POP-1 transgene (*Sallee et al., 2015a*). This view is largely dependent on the notion that high levels of POP-1 correlate to repressive function and that low levels are conducive for activator roles (*Shetty et al., 2005*). In contrast, we did not find evidence of transcriptional activation by POP-1 in the AC/VU precursors nor their differentiated descendants using an established POPTOP (POP-1 and TCF optimal promoter) reporter, which contains seven copies of POP-1/TCF binding sites and the *pes-10* minimal promoter (*Figure 5H*;

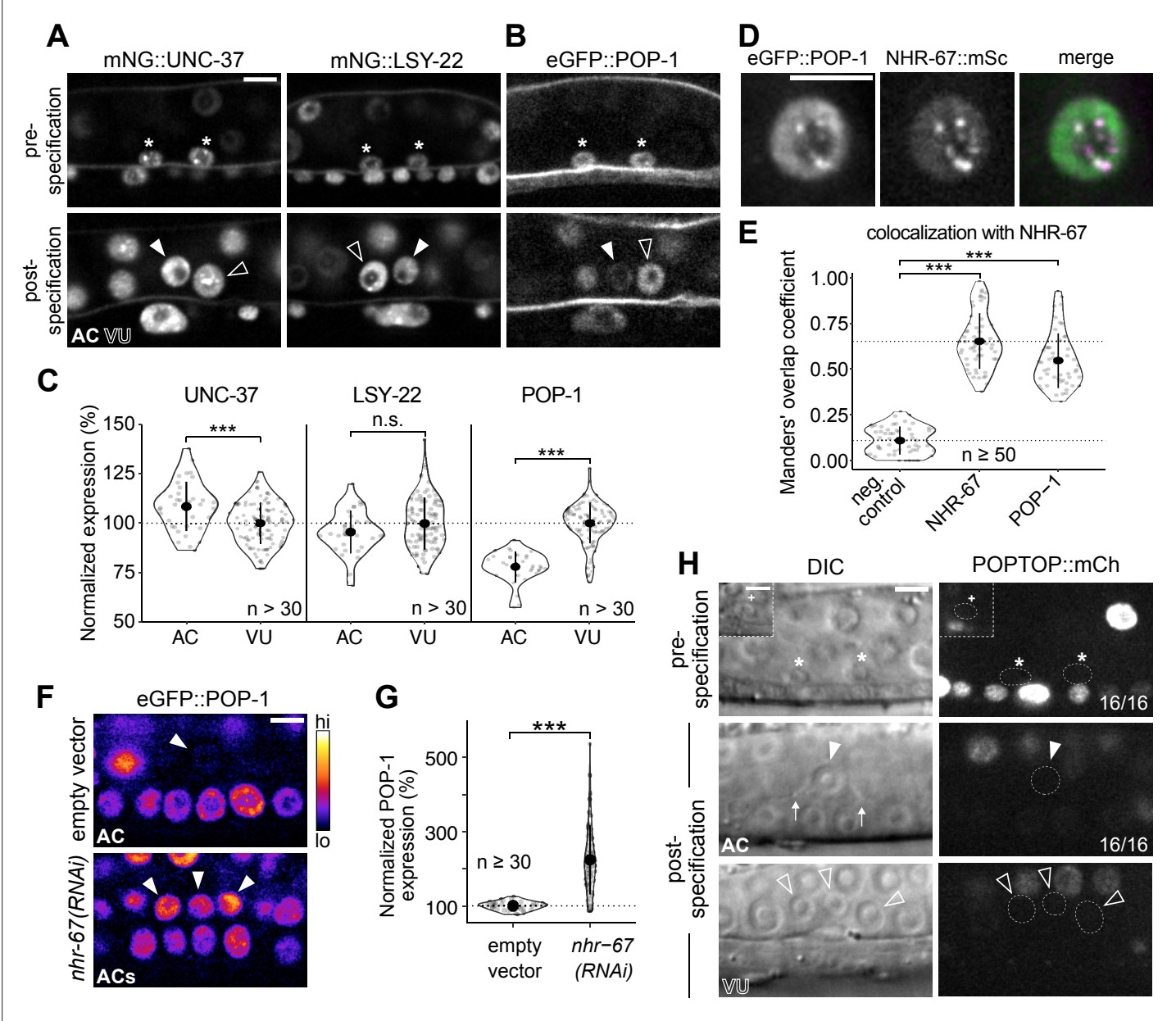

**Figure 5.** POP-1 is enriched in ventral uterine (VU) cells and colocalizes with NHR-67 punctae. (**A–B**) Expression of mNeonGreen::UNC-37 and mNeonGreen::LSY-22 (**A**) and eGFP::POP-1 (**B**) in the anchor cell (AC)/VU precursors pre-specification (left), as well as in the AC and VU cells post-specification (right). (**C**) Quantification of UNC-37, LSY-22, and POP-1 expression at the time of AC invasion. (**D**) Co-visualization of NHR-67::mScarlet-I and eGFP::POP-1 in the VU cells. (**E**) Quantification of POP-1 and NHR-67 colocalization, with plot reporting Manders' overlap coefficient compared to negative and positive controls. (**F-G**) Micrographs (**F**) and quantification (**G**) of eGFP-tagged POP-1 expression in proliferative ACs following RNAi depletion of NHR-67 compared to empty vector control. (**H**) Representative micrographs showing expression of POPTOP, a synthetic *pop-1*-activated reporter construct, in wild-type ACs, VU cells, and their precursors. Insets depict different z-planes of the same image.

The online version of this article includes the following source data and figure supplement(s) for figure 5:

**Source data 1.** Raw data of mNG::UNC-37, mNG::LSY-22, and eGFP::POP-1 expression in the AC/VU precursors, the anchor cell (AC), and ventral uterine (VU) cells, as reported in *Figure 5A–C* and *Figure 5—figure supplement 1A and B*.

**Source data 2.** Raw data of eGFP::POP-1 expression in anchor cells (ACs) resulting from RNAi knockdown of transcription factors and chromatin modifiers compared to empty vector controls, reported in *Figure 5F and G* and *Figure 5—figure supplement 2A and B*.

**Figure supplement 1.** Expression of LSY-22, UNC-37, and POP-1 over developmental time.

**Figure supplement 2.** POP-1 function in ventral uterine (VU) cells is distinct from the activating role in distal somatic gonad.

**Figure supplement 3.** POP-1 expression is regulated by the cell cycle-dependent pro-invasion pathway.

*Figure 5—figure supplement 3A and B*; *Green et al., 2008*). The growing consensus regarding the Wnt/β-catenin asymmetry pathway is that relative levels of POP-1 and β-catenin are more important than absolute protein levels of POP-1 (*Phillips and Kimble, 2009*). Our proposed model of POP-1 acting as a repressor in the proximal gonad is consistent with the finding that SYS-1 (β-catenin) expression is restricted to the distal gonad early in somatic gonad development and is not detectable in the AC or VU cells (*Figure 5—figure supplement 3C*; *Phillips et al., 2007*; *Sallee et al., 2015a*). It is also supported by recent evidence suggesting that UNC-37/LSY-22 mutant alleles phenocopy *pop-1* knockdown, which produces ectopic distal tip cells (*Bekas and Phillips, 2022*).

One aspect that makes studying the repressive role of POP-1 in cell fate maintenance challenging is that its activator function is required for distal cell fate specification in the somatic gonad earlier in development. Loss of either POP-1 and SYS-1 results in a Sys (symmetrical sister cell) phenotype, where all somatic gonad cells adopt the default proximal fate and thereby give rise to ectopic ACs (*Siegfried and Kimble, 2002*; *Siegfried et al., 2004*). This likely occluded previous identification of the potential repressive role of POP-1 in maintaining VU cell fates. To achieve temporal control over POP-1 expression to tease apart its roles, we sought to use the AID system, but the insertion of the degron into the *pop-1* locus disrupted the protein's function. Instead, we paired eGFP-tagged POP-1 with a uterine-specific anti-GFP nanobody (*Smith et al., 2022*; *Wang et al., 2017*). The anti-GFP nanobody is fused to ZIF-1 and serves as an adapter, recognizing GFP-tagged proteins and promoting their ubiquitination by the Cullin2-based E3 ubiquitin ligase, which ultimately targets them for degradation via the proteasome (*Figure 6—figure supplement 1A*; *Wang et al., 2017*). This anti-GFP nanobody, visualized by nuclear expression of mCherry, was not detectable prior to or even shortly after the AC/VU cell fate decision, which allowed us to bypass disruption of initial cell specification (*Figure 6—figure supplement 1B*). While this method only produced a mild knockdown of POP-1 in the VU cells, we still observed the ectopic AC phenotype at low penetrance (7%, n=60) (*Figure 6—figure supplement 1C*). To achieve stronger depletion, we used RNAi for further POP-1 perturbations.

To interrogate the phenotypic consequences of POP-1 perturbation, we utilized a strain expressing two markers of AC fate (*cdh-3p*::mCherry::moeABD and LAG-2::P2A::H2B::mTurquoise2). Following treatment with *pop-1(RNAi)*, we observed several animals with two or more bright *cdh-3/lag-2+* ACs, consistent with known phenotypes caused by cell fate misspecification in the somatic gonad (17%, n=30) (*Figure 6A*). We also observed animals with invasive cells that express AC markers at different levels (53%, n=30), suggesting that the cells did not adopt AC fate at the same time (*Figure 6A*). To test whether the subset of dim *cdh-3/lag-2+* ACs are the result of VU-to-AC cell fate conversion, we visualized AC and VU fates simultaneously using the AC markers previously described along with an mNeonGreen-tagged allele of *lag-1* (CSL), a protein downstream of Notch signaling whose expression becomes restricted to the VU cells following AC/VU cell fate specification. Following treatment with *pop-1(RNAi)*, we found that a subset of ectopic ACs co-express AC markers and LAG-1, likely indicating an intermediate state between the two cell types (*Figure 6—figure supplement 2*). To visualize this process live, we used time-lapse microscopy and were able to capture ectopic ACs gradually upregulating LAG-2 (+51%, n=3) and downregulating LAG-1 (–16%, n=3) over time (*Figure 6B and C*), consistent with VU-to-AC cell fate conversion.

## IDR of NHR-67 facilitates protein-protein interaction with UNC-37

Given that UNC-37, LSY-22, and POP-1 phenocopy each other with respect to AC/VU fates and all three colocalize with NHR-67 punctae, we next sought to further characterize the interactions among these proteins. Previous work has either directly identified or predicted protein-protein interactions among POP-1, UNC-37, and LSY-22 (*Boxem et al., 2008*; *Calvo et al., 2001*; *Flowers et al., 2010*; *Reece-Hoyes et al., 2005*; *Simonis et al., 2009*; *Zhong and Sternberg, 2006*). Using a yeast two-hybrid assay with UNC-37 Gal4-AD prey, we confirmed that UNC-37 directly interacts with both POP-1 and LSY-22 after observing yeast growth on the selective SC-HIS-TRP-LEU plates containing 3-AT (*Figure 7—figure supplement 1*). Using the same technique, we found that NHR-67 binds directly to UNC-37, as previously predicted (*Li et al., 2004*; *Simonis et al., 2009*), but found no evidence of it directly interacting with LSY-22 or POP-1 (*Figure 7—figure supplement 1*).

To further characterize the protein-protein interaction between NHR-67 and UNC-37, we assessed the protein structure of NHR-67 using AlphaFold, an artificial intelligence-based protein structure prediction tool (*Jumper et al., 2021*; *Varadi et al., 2022*), and PONDR, a predictor of intrinsic disorder

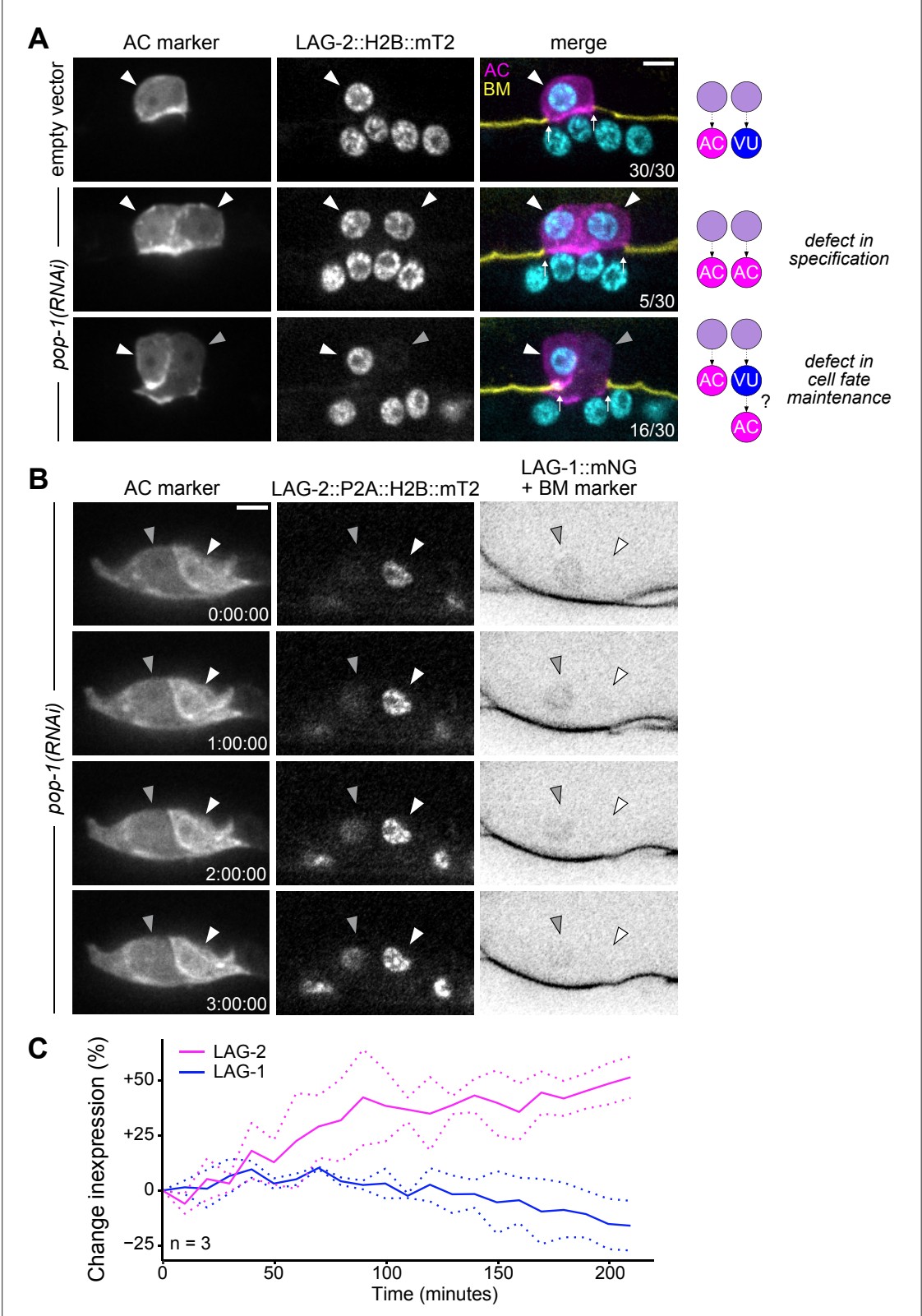

**Figure 6.** Ectopic anchor cells (ACs) arise through VU-to-AC cell fate transformation. (**A**) Representative images of ectopic AC (*cdh-3p*::mCherry::moeABD; LAG-2::P2A::H2B::mTurquoise2) phenotypes observed following RNAi depletion of POP-1. Schematics (right) depict potential explanations for observed phenotypes. (**B**) Expression of AC markers and a VU cell marker (LAG-1::mNeonGreen, inverted to aid visualization) in *pop-*

*Figure 6 continued on next page*

*Figure 6 continued*

1(RNAi) treated animals over time. (**C**) Quantification of LAG-2 (magenta) and LAG-1 (blue) expression in transdifferentiating cells produced by *pop-1(RNAi)* over time.

The online version of this article includes the following source data and figure supplement(s) for figure 6:

**Source data 1.** Raw data of LAG-2::P2A::H2B::mT2 and LAG-1::mNG expression during VU-to-AC transdifferentiation, as reported in *Figure 6B and C*.

**Figure supplement 1.** POP-1 functions to regulate AC/VU cell fates post-specification.

**Figure supplement 2.** Ectopic anchor cells (ACs) resulting from *pop-1* perturbation express ventral uterine (VU) cell markers.

---

(*Peng and Zhang, 2006*). Both identify an intrinsically disordered region (IDR) at the C-terminus of NHR-67 (*Figure 7A and B*). IDRs are low complexity domains that lack fixed three-dimensional structures and have been shown to support dynamic protein-protein interactions (*Chong et al., 2018*). To determine if the IDR of NHR-67 is important for facilitating its interaction with UNC-37, we repeated the yeast two-hybrid experiment using UNC-37 Gal4-AD prey, pairing it with different fragments of the NHR-67 protein: full-length, without its IDR (ΔIDR), and its IDR alone (*Figure 7C and D*). Yeast growth on the selective SC-HIS-TRP-LEU plates containing the competitive inhibitor 3-aminotriazole (3-AT) demonstrates that the 108 amino acid IDR sequence of NHR-67 is necessary and sufficient to bind with UNC-37 (*Figure 7C and D*).

Thus, we propose the following potential model of *C. elegans* uterine cell fate maintenance based on the data presented here and the known roles of Groucho and TCF proteins in regulating transcription. First, transcription of NHR-67 is directly regulated by HLH-2, resulting in its enrichment in the AC compared to the VU cells. In the AC, where NHR-67 levels are high and POP-1 is repressed, NHR-67 is free to activate genes promoting invasive differentiation. In the VU cells, where NHR-67 levels are low and POP-1 levels are high, POP-1 assembles with LSY-22, UNC-37, and NHR-67, either directly repressing NHR-67 targets or sequestering NHR-67 away from its targets (*Figure 7E*). It is possible that POP-1 negatively regulates NHR-67 at the transcriptional level as well, as the NHR-67 promoter contains seven putative TCF binding sites (*Zacharias et al., 2015*).

## Discussion

In summary, here we provide evidence that the activity of the pro-invasive transcription factor, NHR-67, is simultaneously regulated by two distinct processes, which together modulate the proliferative-invasive switch in *C. elegans*. We show that NHR-67 is a potent fate-specifying transcription factor, in that its expression is sufficient for the invasive differentiation of ACs in the somatic gonad. The compartmentalization of NHR-67 in the VU cells could serve as a potential mechanism to suppress its function in activating the pro-invasive program. We also discovered that NHR-67 forms nuclear foci in non-invasive cells, which exhibit liquid-like properties, indicated by observations of their condensation, dissolution, and relatively rapid recovery from photobleaching, similar to what has been described with P granules (*Brangwynne et al., 2009*). These NHR-67 punctae associate with Groucho homologs, UNC-37 and LSY-22, likely through a direct protein-protein interaction with UNC-37 mediated by the C-terminal IDR of NHR-67. We postulate that this association leads to protein condensation, as has recently been described in *Ciona* embryos (*Treen et al., 2021*). Furthermore, repression of the default invasive state appears to be dependent on the expression of the TCF/LEF homolog POP-1, which clarifies our understanding of the dual roles this protein plays during the development of the somatic gonad. It is also interesting to note that the dynamic punctae formed by POP-1 in non-Wnt signaled cells was first described 20 years ago (*Maduro et al., 2002*), but their function is only now being appreciated in light of recent advances in our understanding of the formation of higher-order associations in the nucleus.

With regard to protein compartmentalization in the nucleus, most research has been through the lens of transcriptional activation through RNA Polymerase II and the mediator complex (*Boija et al., 2018*; *Cho et al., 2018*; *Sabari et al., 2018*) or repression through HP1 heterochromatin proteins (*Larson et al., 2017*; *Strom et al., 2017*). Here, we report the second observed case of compartmentalization of Groucho proteins (*Treen et al., 2021*), which may suggest that Groucho proteins have evolutionarily conserved roles that require this type of subnuclear organization.

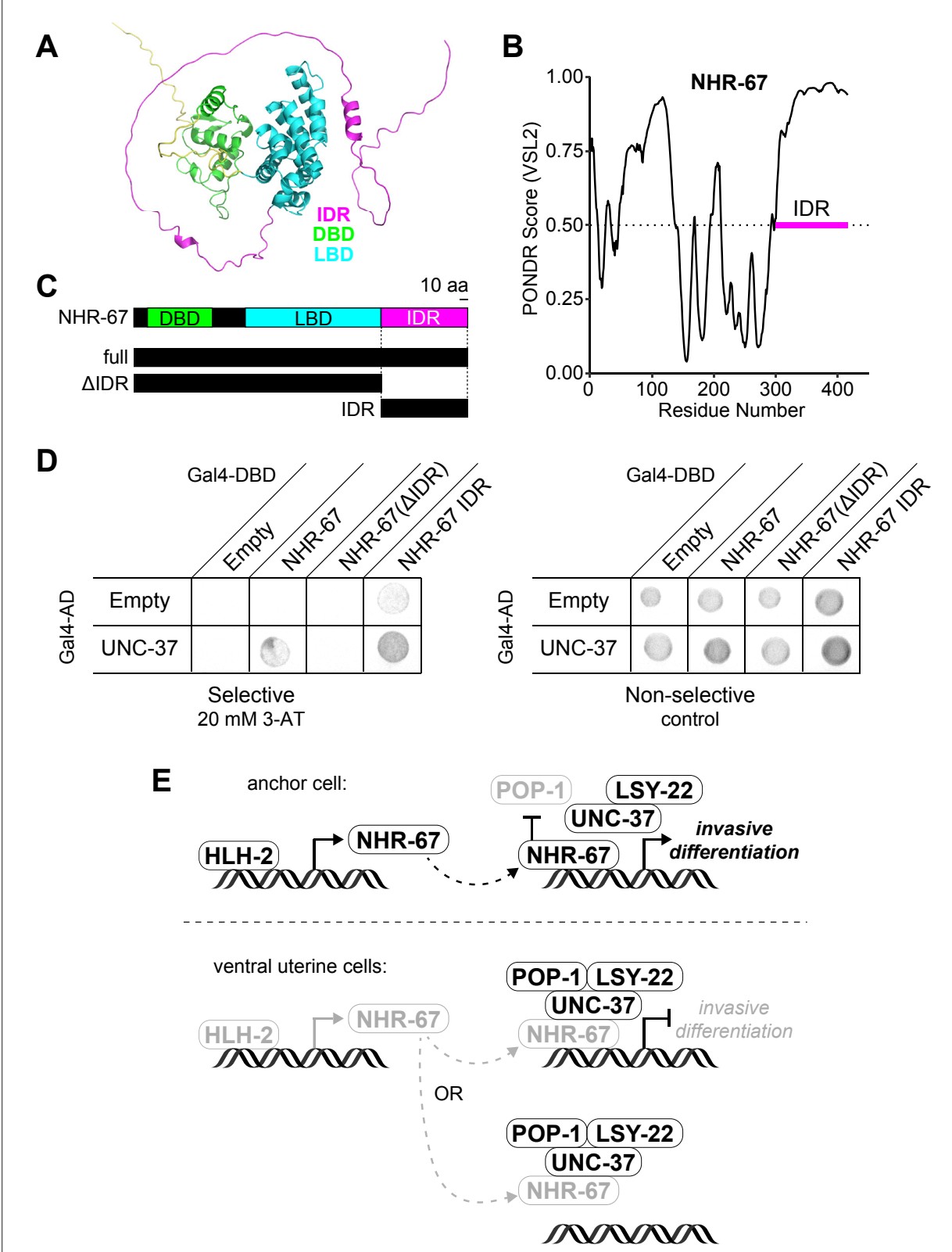

**Figure 7.** NHR-67 binds to UNC-37 through IDR-mediated protein-protein interaction. (**A**) Predicted structure of NHR-67 generated by AlphaFold. (**B**) Measure of intrinsic disorder of NHR-67 using the PONDR VSL2 prediction algorithm. (**C**) Schematic of NHR-67 protein-coding sequences used for Yeast two-hybrid experiments with reference to its intrinsically disordered region (IDR, magenta), DNA binding domain (DBD, green), and ligand binding domain (LBD, cyan). Scale bar, 10 amino acids. (**D**) Yeast two-hybrid experiment shows pairing of UNC-37 with either full-length NHR-67 or the IDR alone

*Figure 7 continued on next page*

*Figure 7 continued*

allows for yeast growth in the presence of competitive inhibitor 3-AT (20 mM). (**E**) Possible models of the roles of NHR-67, UNC-37, LSY-22, and POP-1 in the maintenance of anchor cell (AC) and ventral uterine (VU) cell fate. In the ventral uterine cells, the association of NHR-67 with the Groucho/TCF complex may result in the repression of NHR-67 targets (top) or the sequestration of NHR-67 away from its targets (bottom).

The online version of this article includes the following figure supplement(s) for figure 7:

**Figure supplement 1.** NHR-67 exhibits protein-protein interaction with UNC-37.

Still, as this is one of the first studies into the compartmentalization of transcriptional repressors in vivo, there is much left to learn. For example, it is unknown whether DNA binding is necessary for nuclear puncta formation. The interaction between UNC-37 and NHR-67 does not appear to depend on DNA binding, as the C-terminal IDR region of NHR-67 (excluding its zinc finger domains) was sufficient for binding with UNC-37 in vitro, but it is possible that DNA binding is needed for oligomerization in vivo. Furthermore, it remains unclear if suppression of invasive differentiation is achieved by simply sequestering the pro-invasive transcription factor NHR-67 away from its transcriptional targets or through direct repression of transcription. If the latter, another question that arises is how the repressive complex gets recruited to specific genomic sites, since POP-1 and NHR-67 are both capable of binding to DNA, and whether repression is achieved through competition with transcriptional activators or recruitment of histone deacetylases. Direct targets of NHR-67 have not yet been discovered, which makes it difficult to investigate this specific aspect of the repressive mechanism at present. We see this as a promising avenue of future study as technologies advance, allowing for transcriptional profiling and target identification in specific tissues or cells (*Gómez-Saldivar et al., 2020*; *Katsanos and Barkoulas, 2022*).

In this work, we have also identified several perturbations (i.e. increasing levels of NHR-67, decreasing levels of UNC-37/LSY-22) that result in incompletely penetrant transdifferentiation phenotypes and/or intermediate cell fates. We foresee these being ideal cell fate challenge backgrounds in which to perform screens to identify regulators of cellular plasticity, as has been done in other contexts (*Rahe and Hobert, 2019*). Additionally, these induced fate transformations can be paired with tools to visualize and manipulate the cell cycle (*Adikes et al., 2020*) to determine if any cell cycle state is particularly permissive for cell fate plasticity. While G1 arrest has been shown to enhance the conversion of human fibroblasts to dopaminergic neurons (*Jiang et al., 2015*), mitosis is required for the natural K-to-DVB transdifferentiation event in *C. elegans* (*Riva et al., 2022*). As control of proliferation and invasion, as well as maintenance of differentiated cellular identities, are important for both homeostatic and disease states, it is our hope that this work will shed light on how cells switch between these states in the context of cancer growth and metastasis.

# Materials and methods

## *C. elegans* strains, culture, and nomenclature

Methods for *C. elegans* culture and genetics were followed as previously described (*Brenner, 1974*). Developmental synchronization for experiments was achieved through alkaline hypochlorite treatment of gravid adults to isolate eggs (*Porta-de-la-Riva et al., 2012*). L1 stage animals were plated on nematode growth media plates and subsequently cultured at 20 °C or 25 °C. Heat shock-inducible transgenes were activated by incubating animals on plates sealed with Parafilm in a 33 °C water bath for 2–3 hr. In the text and in figures, promoter sequences are designated with a '*p*' following the gene name and gene fusions are represented by a double-colon (::) symbol.

## CRISPR/Cas9 injections

New alleles and single-copy transgenes were generated by homology-directed repair using CRISPR-based genome engineering. mScarlet::AID and mNeonGreen::AID were inserted into the C-terminus of the NHR-67 locus by injecting adult germlines with Cas9 guide-RNA ribonucleoprotein complexes and short single-stranded oligodeoxynucleotide donors, as previously described (*Ghanta and Mello, 2020*). Successful integration was identified through screening for fluorescence and PCR. The LSY-22 locus was edited by injecting a Cas9 guide RNA plasmid and repair template plasmid containing a self-excising cassette with selectable markers to facilitate screening (*Dickinson et al., 2015*; *Dickinson*

*and Goldstein, 2016*; *Huang et al., 2021*). Repair templates used to tag LSY-22 with TagRFP-T::AID and mNeonGreen::AID were generated by cloning ~750–850 bp homology arms into pTNM063 and pDD312, respectively (*Hearn et al., 2021*; *Dickinson et al., 2015*). All guide and repair sequences used can be found in *Supplementary file 1*.

## Existing alleles

The GFP-tagged alleles of the pro-invasive transcription factors (*egl-43, fos-1, hlh-2,* and *nhr-67*) and the TagRFP-T::AID-tagged NHR-67 allele were generated in preceding work (*Medwig-Kinney et al., 2021*; *Medwig-Kinney et al., 2020*). Recent micropublications describe the P2A::H2B::mTurquoise2-tagged *lag-2* and mNeonGreen-tagged *lin-12* alleles used in this study (*Medwig-Kinney et al., 2022*; *Pani et al., 2022*). The eGFP-tagged *pop-1* allele and POPTOP reporter were previously published (*Green et al., 2008*; *van der Horst et al., 2019*), as were the AID::BFP and mNeonGreen tagged alleles of *unc-37* (*Kurashina et al., 2021*; *Ma et al., 2021*). GFP-tagged *ama-1* (*Hills-Muckey et al., 2022*) as well as mKate2-tagged *hpl-1* and *hpl-2* (*Patel and Hobert, 2017*) were also disseminated in prior publications. The single-copy transgenes expressing the CDK sensor and TIR1 variants under ubiquitously expressed ribosomal promoters (*rps-27* and *rpl-28*, respectively) as well as the tissue-specific GFP-targeting nanobody are described in previous work (*Adikes et al., 2020*; *Hills-Muckey et al., 2022*; *Smith et al., 2022*; *Wang et al., 2017*) and are located at neutral genomic sites, ttTi4348 or ttTi5605 (*Frøkjær-Jensen et al., 2013*). The same is true for the heat shock inducible constructs for HLH-2 and NHR-67 (*Medwig-Kinney et al., 2020*). The cadherin (*cdh-3*) anchor cell reporter and basement membrane (laminin) markers have already been characterized (*Keeley et al., 2020*; *Matus et al., 2010*). The following mutant alleles were obtained from the *Caenorhabditis* Genetics Center: *unc-37(e262wd26)* and *unc-37(wd17wd22)* (*Pflugrad et al., 1997*), the latter of which was maintained using the chromosome I/III balancer *hT2* (*McKim et al., 1993*). The genotypes of all strains used in this study can be found in the Key Resources Table.

## Auxin inducible protein degradation

The auxin-inducible degron (AID) system was used for the strong depletion of proteins of interest (*Zhang et al., 2015*). AID-tagged alleles were paired with the *Arabidopsis thaliana* F-box protein, transport inhibitor response 1 ($_{At}$TIR1), and treated with the water-soluble auxin 1-Naphthaleneacetic acid (K-NAA) at 1 mM concentration (*Martinez et al., 2020*). Auxin was added to nematode growth media plates according to previously published protocols (*Martinez and Matus, 2020*), which were then seeded with OP50 *E. coli*. To achieve robust depletion, synchronized L1 stage animals were directly plated on auxin plates.

## RNA interference

The RNAi clones targeting *pop-1* and *uba-1* and the corresponding empty vector control (L4440) were obtained from the Vidal library (*Rual et al., 2004*). The RNAi constructs targeting the pro-invasive transcription factors (*egl-43, fos-1, hlh-2,* and *nhr-67*) and chromatin modifiers (*pbrm-1, swsn-4,* and *swsn-8*) are derived from the highly efficient RNAi vector T444T (*Sturm et al., 2018*) and were generated in preceding work (*Medwig-Kinney et al., 2020*; *Smith et al., 2022*). To avoid known AC/VU cell fate specification defects caused by *hlh-2* perturbations, synchronized animals were grown on OP50 until the L2 stage when they were then shifted to *hlh-2* RNAi plates.

## Live-cell imaging

With the exception of the FRAP experiments shown in *Figure 3*, all micrographs were collected on a Hamamatsu Orca EM-CCD camera mounted on an upright Zeiss AxioImager A2 with a Borealis-modified CSU10 Yokagawa spinning disk scan head (Nobska Imaging) using 405 nm, 440 nm, 488 nm, 514 nm, and 561 nm Vortran lasers in a VersaLase merge and a Plan-Apochromat 100x/1.4 (NA) Oil DIC objective. MetaMorph software (Molecular Devices) was used for microscopy automation. Several experiments were scored using epifluorescence visualized on a Zeiss Axiocam MRM camera, also mounted on an upright Zeiss AxioImager A2 and a Plan-Apochromat 100x/1.4 (NA) Oil DIC objective. For static imaging, animals were mounted into a drop of M9 on a 5% Noble agar pad containing approximately 10 mM sodium azide anesthetic and topped with a coverslip. For long-term time-lapse imaging, animals were first anesthetized in 5 mM levamisole diluted in M9 for approximately

20 min, then transferred to a 5% Noble agar pad and topped with a coverslip sealed with VALAP (*Kelley et al., 2017*). For short-term time-lapse imaging, the pre-anesthetization step was omitted, and animals were transferred directly into a drop of 5 mM levamisole solution on the slide.

## Fluorescence recovery after photobleaching

FRAP experiments were performed using an Acal BFi UV Optimicroscan photostimulation device mounted on a spinning disk confocal system consisting of a Nikon Ti2 inverted microscope with Yokogawa CSU-W1 SoRa spinning disk. Data were acquired using a Hamamatsu ORCA Fusion camera, 60x 1.27 NA water immersion objection, SoRa disk, and 2.8 x SoRa magnifier. Single plane images were collected every 1 s.

## Yeast one-hybrid

The 276 bp fragment of the NHR-67 promoter (*Bodofsky et al., 2018*) was cloned into the pMW2 vector, and linearized by BamHI digestion. Linearized plasmid was transformed into the Y1H yeast strain (as described in *Reece-Hoyes and Walhout, 2018*). Transformed yeast was plated on SC-HIS plates for three days before being transformed with the HLH-2 Gal4-AD plasmid. Three colonies from each transformation plate were streaked onto SC-HIS-TRP+3-aminotriazole (3-AT) plates. Protein-DNA interactions were determined by visible growth on 3-AT conditions with negative growth in empty vector controls after three days. Plates were imaged on a Fotodyne FOTO/Analyst Investigator/FX darkroom imaging station.

## Yeast two-hybrid

Plasmids containing target proteins fused to GAL-4 DNA-binding-domain + LEU and GAL-4 Activation Domain + TRP were co-transformed into the pJ69-4a Y2H yeast strain as previously described (*Reece-Hoyes and Walhout, 2018*). Transformed yeast was plated on SC-TRP-LEU plates for three days. Three colonies from each transformation plate were streaked onto SC-HIS-TRP-LEU 3-AT plates. Protein interactions were determined by visible growth on 3-AT conditions with negative growth in empty vector controls after three days. Plates were imaged as described in the previous section.

## Quantification of protein expression and cell cycle state

Image quantification was performed in Fiji/ImageJ (*Schindelin et al., 2012*). Protein expression was quantified by drawing a region of interest and measuring the mean gray value, then manually subtracting the mean gray value of a background region of similar area to account for camera noise. For nuclear-localized proteins, the region of interest was drawn around the nucleus. Membrane expression of LIN-12 was measured by drawing a line (1.5 pixel thickness) along a cell's basolateral surface, to distinguish between LIN-12 expression from the cell of interest and its neighbors. CDH-3 expression was measured by drawing a region of interest around the cell membrane excluding the nucleus (as the nuclear-localized TIR1 transgene was expressed in the same channel). The CDK sensor was quantified as previously described (*Adikes et al., 2020*). Following rolling ball subtraction (50 pixels), the mean gray value is measured in a region of interest drawn within the cytoplasm and one around the nucleus excluding the nucleolus. The cytoplasmic-to-nuclear ratio correlates to CDK activity and is used to assess the cell cycle state (*Adikes et al., 2020*; *Spencer et al., 2013*). Movies were collected by acquiring z-stacks at 5 min intervals. Samples were time-aligned relative to anaphase. Cells that did not undergo anaphase during the acquisition period were aligned based on their DHB ratios. Animals that were arrested in development (i.e. did not show evidence of progressing through the cell cycle) were excluded from the analysis.

## Colocalization analyses

For colocalization analyses, single-plane images were collected to avoid z drift during acquisition and prevent photobleaching, which was often non-uniform between red and green fluorophores. Micrographs were subject to background subtraction (rolling ball radius = 50) followed by thresholding to segment punctae. Manders' overlap coefficients (M) were calculated by measuring the extent that segmented punctae of NHR-67 overlapped with that of other proteins using Just Another Colocalization Plugin (JACoP) in Fiji/ImageJ (*Bolte and Cordelières, 2006*; *Schindelin et al., 2012*). Heterozygous animals for *nhr-67*::mScarlet and *nhr-67*::GFP were used as positive controls. These images were

then re-analyzed following a 90-degree rotation of one of the two channels being compared, resulting in random colocalization that served as a negative control.

## Data visualization and statistical analyses

Representative images were processed using Fiji/ImageJ (*Schindelin et al., 2012*). Heat maps were generated using the Fire lookup table. A power analysis was performed prior to data collection to determine sample sizes (*Cohen, 1992*). Tests to determine the statistical significance of data were conducted in RStudio and plots were generated using the R package ggplot2 (*Wickham, 2016*). Error bars represent the mean ± standard deviation. Schematics of gene loci were generated using sequences from WormBase (*Harris et al., 2020*) and the Exon-Intron Graphic Maker (http://wormweb.org/exonintron). Figures were assembled in Adobe Illustrator.

# Acknowledgements

We are grateful to Dr. Derek Applewhite and Aidan Teran for advice on the quantification of protein colocalization. Additionally, we thank Chris Zhao for constructive comments on the manuscript. Some strains were provided by the *Caenorhabditis* Genetics Center, which is funded by the NIH Office of Research Infrastructure Programs (P40 OD010440).

# Additional information

### Competing interests

Neha Somineni: Paid employee of Integra LifeSciences. David Q Matus: Paid employee of Arcadia. The other authors declare that no competing interests exist.

### Funding

| Funder | Grant reference number | Author |
| --- | --- | --- |
| National Institutes of Health | R01GM121597 | David Q Matus |
| Damon Runyon Cancer Research Foundation | DRR-47-17 | David Q Matus |
| National Institutes of Health | F31HD100091 | Taylor N Medwig-Kinney |
| Stony Brook University | Presidential Critical Research Funds | Taylor N Medwig-Kinney |
| National Institutes of Health | F30CA257383 | Michael AQ Martinez |
| Human Frontier Science Program | LTF000127/2016-L | Callista Yee |
| Howard Hughes Medical Institute | Investigator | Kang Shen |
| National Institutes of Health | R01GM117406 | Christopher Hammell |
| National Science Foundation | 2217560 | Christopher Hammell |
| National Institutes of Health | R35GM142880 | Ariel M Pani |

The funders had no role in study design, data collection and interpretation, or the decision to submit the work for publication.

### Author contributions

Taylor N Medwig-Kinney, Conceptualization, Formal analysis, Supervision, Funding acquisition, Investigation, Visualization, Writing – original draft; Brian A Kinney, Ariel M Pani, Investigation, Writing

– review and editing; Michael AQ Martinez, Resources, Investigation, Writing – review and editing; Callista Yee, Resources, Writing – review and editing; Sydney S Sirota, Resources, Formal analysis, Investigation; Angelina A Mullarkey, Formal analysis, Writing – review and editing; Neha Somineni, Justin Hippler, Resources, Formal analysis; Wan Zhang, Resources; Kang Shen, Supervision; Christopher Hammell, Resources, Supervision, Writing – review and editing; David Q Matus, Conceptualization, Supervision, Funding acquisition, Writing – review and editing

**Author ORCIDs**
Taylor N Medwig-Kinney (iD) https://orcid.org/0000-0001-7989-3291
Brian A Kinney (iD) https://orcid.org/0000-0001-5628-1436
Michael AQ Martinez (iD) https://orcid.org/0000-0003-1178-7139
Callista Yee (iD) http://orcid.org/0000-0002-2928-492X
Angelina A Mullarkey (iD) http://orcid.org/0000-0002-5830-5347
Neha Somineni (iD) https://orcid.org/0000-0001-5702-1695
Justin Hippler (iD) https://orcid.org/0000-0002-7026-8761
Kang Shen (iD) http://orcid.org/0000-0003-4059-8249
Christopher Hammell (iD) http://orcid.org/0000-0002-5961-0976
David Q Matus (iD) https://orcid.org/0000-0002-1570-5025

Reviewer #1 (Public Review): https://doi.org/10.7554/eLife.84355.3.sa1
Reviewer #2 (Public Review): https://doi.org/10.7554/eLife.84355.3.sa2
Author Response https://doi.org/10.7554/eLife.84355.3.sa3

## Additional files

**Supplementary files**
• MDAR checklist
• Supplementary file 1. Sequences used in this study.

**Data availability**
All data generated or analyzed during this study are included in the manuscript and supporting files. New strains that have not been deposited with the Caenorhabditis Genetics Center will be made available upon request.

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

## Appendix 1

### Appendix 1—key resources table

| Reagent type (species) or resource | Designation | Source or reference | Identifiers | Additional information |
|---|---|---|---|---|
| Strain, strain background (*C. elegans*) | DQM335 | *Medwig-Kinney et al., 2020* | | *egl-43(bmd88[egl-43p::EGL-43::loxP::GFP::EGL-43]) II; qyIs225[cdh-3p::mCherry::moeABD] V; qyIs7[laminin::GFP] X.* |
| Strain, strain background (*C. elegans*) | DQM350 | *Medwig-Kinney et al., 2020* | | *hlh-2(bmd90[hlh-2p::loxP::GFP::HLH-2]) I; qyIs225[cdh-3p::mCherry::moeABD] V; qyIs7[laminin::GFP] X.* |
| Strain, strain background (*C. elegans*) | DQM354 | This paper | | *nhr-67(syb509[nhr-67p::NHR-67::GFP]) IV; bmd66[loxP::egl-43p::GFP-nanobody::P2A::HIS-58::mCherry] I; qyIs225[cdh-3p::mCherry::moeABD] V; qyIs7[laminin::GFP] X.* |
| Strain, strain background (*C. elegans*) | DQM368 | *Medwig-Kinney et al., 2020* | | *nhr-67(syb509[nhr-67p::NHR-67::GFP]) IV; qyIs225[cdh-3p::mCherry::moeABD] V; qyIs7[laminin::GFP] X.* |
| Strain, strain background (*C. elegans*) | DQM444 | *Medwig-Kinney et al., 2020* | | *bmd121[hsp::NHR-67::2x-BFP] I; qyIs227[cdh-3p::mCherry::moeABD] I; qyIs7[laminin::GFP] X.* |
| Strain, strain background (*C. elegans*) | DQM515 | *Medwig-Kinney et al., 2020* | | *fos-1(bmd138[fos-1p::loxP::GFP::FOS-1]) V; qyIs227[cdh-3p::mCherry::moeABD] I; qyIs7[laminin::GFP] X.* |
| Strain, strain background (*C. elegans*) | DQM704 | *Medwig-Kinney et al., 2021* | | *nhr-67(bmd212[nhr-67p::NHR-67::TagRFP-T::AID]) IV; hlh-2(bmd90[hlh-2p::LoxP::GFP::HLH-2]) I.* |
| Strain, strain background (*C. elegans*) | DQM800 | This paper | | *pop-1(he335[pop-1p::eGFP::loxP::POP-1]) I; syIs187[pes-10::7XTCF-mCherry-let-858(3'UTR)+unc-119(+)].* |
| Strain, strain background (*C. elegans*) | DQM811 | This paper | | *qyIs227[cdh-3p::mCherry::moeABD] I; lam-2(qy20[lam-2p::LAM-2::mNeonGreen]) X; lag-2(bmd202[lag-2p::LAG-2::P2A::H2B::mTurquoise2^lox511^ 2xHA]) V.* |
| Strain, strain background (*C. elegans*) | DQM853 | This paper | | *hlh-2(bmd90[hlh-2p::loxP::GFP::HLH-2]) I; stIs11476[nhr-67p::NHR-67::H1-wCherry+unc-119(+)].* |
| Strain, strain background (*C. elegans*) | DQM957 | This paper | | *csh128[rpl-28p::TIR1::T2A::mCherry::his-11] II; qyIs225[cdh-3p:: mCherry::moeABD] V; qyIs7[laminin::GFP] X.* |
| Strain, strain background (*C. elegans*) | DQM958 | This paper | | *csh140[rpl-28p::TIR1(F79G)::T2A::mCherry::his-11] II; qyIs225[cdh-3p:: mCherry::moeABD] V; qyIs7[laminin::GFP] X.* |
| Strain, strain background (*C. elegans*) | DQM971 | This paper | | *pop-1(he335[pop-1p::eGFP::loxP::POP-1]) I; qyIs225[cdh-3p::mCherry::moeABD] V; qyIs7[laminin::GFP] X.* |
| Strain, strain background (*C. elegans*) | DQM989 | This paper | | *unc-37(devKi218[unc-37p::mNeonGreen::UNC-37]) I; qyIs225[cdh-3p::mCherry::moeABD] V; qyIs7[laminin::GFP] X.* |
| Strain, strain background (*C. elegans*) | DQM990 | This paper | | *unc-37(e262wd26) I; qyIs225[cdh-3p::mCherry::moeABD] V; qyIs7[laminin::GFP] X.* |
| Strain, strain background (*C. elegans*) | DQM1003 | This paper | | *nhr-67(syb509[nhr-67p::NHR-67::GFP]) IV; bmd168[rps-27p::DHB::2x-mKate2] II.* |
| Strain, strain background (*C. elegans*) | DQM1006 | This paper | | *LSY-22(bmd275[lsy-22p::loxP::mNeonGreen::AID::LSY-22]) I; qyIs225[cdh-3p::mCherry::moeABD] V; qyIs7[laminin::GFP] X.* |

*Appendix 1 Continued on next page*

*Appendix 1 Continued*

| Reagent type (species) or resource | Designation | Source or reference | Identifiers | Additional information |
|---|---|---|---|---|
| Strain, strain background (*C. elegans*) | DQM1008 | This paper | | pop-1(he335[pop-1p::eGFP::loxP::POP-1]) I; bmd277[loxP::egl-43p::GFP-nanobody::P2A::HIS-58::mCherry] I; qyIs225[cdh-3p::mCherry::moeABD] V; qyIs7[laminin::GFP] X. |
| Strain, strain background (*C. elegans*) | DQM1009 | This paper | | unc-37(devKi218[unc-37p::mNeonGreen::UNC-37]) I; nhr-67(wy1633[nhr-67p::NHR-67::mScarlet-I::AID*::3xFLAG]) IV. |
| Strain, strain background (*C. elegans*) | DQM1010 | This paper | | hpl-2(ot860[hpl-2p::HPL-2::mKate2::HPL-2]) III; nhr-67(syb509[nhr-67p::NHR-67::GFP]) IV. |
| Strain, strain background (*C. elegans*) | DQM1011 | This paper | | hpl-1(ot841[hpl-1p::HPL-1::mKate2::HPL-1]) X; nhr-67(syb509[nhr-67p::NHR-67::GFP]) IV. |
| Strain, strain background (*C. elegans*) | DQM1012 | This paper | | LSY-22(bmd214[lsy-22p::lox2272::TagRFP-T::AID::LSY-22]) I; nhr-67(syb509[nhr-67p::NHR-67::GFP]) IV. |
| Strain, strain background (*C. elegans*) | DQM1013 | This paper | | pop-1(he335[pop-1p::eGFP::loxP::POP-1]) I; nhr-67(syb509[nhr-67p::NHR-67::GFP]) IV. |
| Strain, strain background (*C. elegans*) | DQM1014 | This paper | | unc-37(wd17wd22)/hT2[bli-4(e937) let-?(q782) qIs48] (I, III); qyIs225[cdh-3p::mCherry::moeABD] V; qyIs7[laminin::GFP] X. |
| Strain, strain background (*C. elegans*) | DQM1017 | This paper | | ama-1(ers49[ama-1p::AMA-1::AID::GFP]) IV; nhr-67(wy1633[nhr-67p::NHR-67::mScarlet-I::AID*::3xFLAG]) IV. |
| Strain, strain background (*C. elegans*) | DQM1051 | This paper | | lin-12(ljf31[lin-12::mNeonGreen[C1]^loxP^3xFlag]) III; lag-2(bmd202[lag-2p::LAG-2::P2A::H2B::mTurquoise2^lox511^ 2xHA]) V. |
| Strain, strain background (*C. elegans*) | DQM1081 | This paper | | bmd168[rps-27p::DHB::2x-mKate2] II; egl-13(devKi199[egl-13p::EGL-13::mNeonGreen]) X; lag-2(bmd202[lag-2p::LAG-2::P2A::H2B::mTurquoise2]) V. |
| Strain, strain background (*C. elegans*) | DQM1101 | This paper | | lsy-22(bmd275[lsy-22p::^loxP^mNeonGreen::AID::LSY-22]) I; csh128[rpl-28p::TIR1::P2A::mCherry::his-11] II; qyIs225[cdh-3p:: mCherry::moeABD] V; qyIs7[laminin::GFP] X. |
| Strain, strain background (*C. elegans*) | DQM1115 | This paper | | unc-37(miz36[unc-37p::UNC-37::AID::BFP]) I; csh128[rpl-28p::TIR1::P2A::mCherry::his-11] II; qyIs225[cdh-3p:: mCherry::moeABD] V; qyIs7[laminin::GFP] X. |
| Strain, strain background (*C. elegans*) | DQM1127 | This paper | | nhr-67(syb509[nhr-67p::NHR-67::GFP]) IV; stIs11476[nhr-67p::NHR-67::H1-wCherry+unc-119(+)]. |
| Strain, strain background (*C. elegans*) | DQM1129 | This paper | | bmd143[hsp::HLH-2::2xBFP] I; nhr-67(syb509[nhr-67p::NHR-67::GFP]) IV. |
| Strain, strain background (*C. elegans*) | DQM1135 | This paper | | qyIs227[cdh-3p::mCherry::moeABD] I; lam-2(qy20[lam-2p::LAM-2::mNeonGreen]) X; lag-2(bmd202[lag-2p::LAG-2::P2A::H2B::mTurquoise2^lox511^ 2xHA]) V; lag-1(devKi208[lag-1::mNeonGreen]) IV. |
| Strain, strain background (*C. elegans*) | JK3791 | *Phillips et al., 2007* | | qIs95[sys-1p::Venus::SYS-1+pttx-3::DsRed] |
| Strain, strain background (*C. elegans*) | NK1034 | *Matus et al., 2015* | | qyIs225[cdh-3p::mCherry::moeABD] V; qyIs7[laminin::GFP] X. |
| Strain, strain background (*C. elegans*) | PHX509 | *Medwig-Kinney et al., 2020* | | nhr-67(syb509[nhr-67p::NHR-67::GFP]) IV. |

*Appendix 1 Continued on next page*

*Appendix 1 Continued*

| Reagent type (species) or resource | Designation | Source or reference | Identifiers | Additional information |
|---|---|---|---|---|
| Strain, strain background (*C. elegans*) | PS5332 | *Green et al., 2008* | | *syls187[pes-10::7XTCF-mCherry-let-858(3'UTR)+unc-119(+)]* |
| Strain, strain background (*C. elegans*) | RW11476 | *Gerstein et al., 2010* | | *unc-119(tm4063) III; stIs11476[nhr-67::H1-wCherry+unc-119(+)].* |
| Strain, strain background (*C. elegans*) | SV2114 | *van der Horst et al., 2019* | | *pop-1(he335[eGFP::loxP::pop-1]) I.* |
| Strain, strain background (*C. elegans*) | TV27467 | This paper | | *nhr-67(wy1632[nhr-67p::NHR-67::mNeonGreen::AID*::3xFLAG]) IV.* |
| Strain, strain background (*C. elegans*) | TV27468 | This paper | | *nhr-67(wy1633[nhr-67p::NHR-67::mScarlet-I::AID*::3xFLAG]) IV.* |
| Recombinant DNA reagent | Plasmid: pTNM087 | This paper | | *LSY-22 sgRNA plasmid (AAACGAAGTGGATCAGCCAG)* |
| Recombinant DNA reagent | Plasmid: pTNM088 | This paper | | *LSY-22^SEC^TagRFP-T::AID repair plasmid* |
| Recombinant DNA reagent | Plasmid: pTNM140 | This paper | | *LSY-22^SEC^mNeonGreen::AID repair plasmid* |
| Chemical compound, drug | 1-Naphthaleneacetic acid, potassium salt (K-NAA) | PhytoTech Labs | N610 | |
| Chemical compound, drug | Hygromycin B | Omega Scientific, Inc. | HG-80 | |
| Chemical compound, drug | Levamisole hydrochloride | Sigma-Aldrich | 31742 | |
| Chemical compound, drug | Sodium azide | Sigma-Aldrich | S2002 | |
| Software, algorithm | Adobe Illustrator | Adobe | Version 26.0.2 | |
| Software, algorithm | Alpha Fold | *Jumper et al., 2021*; *Varadi et al., 2022* | Version 2 | |
| Software, algorithm | ApE – A Plasmid Editor | Wayne Davis | Version 2.0.61 | |
| Software, algorithm | Fiji/ImageJ | *Schindelin et al., 2012* | Version 2.0.0-rc-69/1.53e | |
| Software, algorithm | ggplot2 | Tidyverse | Version 3.3.5 | |
| Software, algorithm | Exon-Intron Graphic Maker | Nikhil Bhatla | Version 4 | |
| Software, algorithm | JACoP (Just Another Colocalization Plugin) | *Bolte and Cordelières, 2006* | Version 2.1.1 | |
| Software, algorithm | Metamorph | Molecular Devices | Version 7.10.3.279 | |
| Software, algorithm | Rstudio | R | Version 1.4.1717 | |

