## [Editor Report · eLife assessment]

This **valuable** data study presents **convincing** data that expression of the *C. elegans* transcription factor NHR-67 is sufficient to drive an invasive fate, and that the alternative proliferative fate is associated with NHR-67 transcriptional down-regulation. While the observation that NHR-67 forms punctae associated with transcriptional repressors in non-invasive cells is intriguing, the work does not yet established a clear link between the formation and dissolution of NHR-67 condensates with the activation of downstream genes that NHR-67 is actively repressing. The work will be of interest to developmental biologists studying transcriptional control of cell fate specification in animals, especially once issues around the functional significance of the NHR-67 containing punctae are resolved.

---

## [Referee Report · Reviewer #1 (Public Review)]

Medwig-Kinney et al perform the latest in a series of studies unraveling the genetic and physical mechanisms involved in the formation of *C. elegans* gonad. They have paid particular attention to how two different cell fates are specified, the ventral uterine (VU) or anchor cell (AC), and the behaviors of these two cell types. This cell fate choice is interesting because the anchor cell performs an invasive migration through a basement membrane. A process that is required for correct *C. elegans* gonad formation and that can act as a model for other invasive processes, such as malignant cancer progression. The authors have identified a range of genes that are involved in the AC/VC fate choice, and that impart the AC cell with its ability to arrest the cell cycle and perform an invasive migration. Taking advantage of a range of genetic tools, the authors show that the transcription factor NHR-63 is strongly expressed in the AC cell. The authors also present evidence that NHR-63 is could function as a transcriptional repressor through interactions with a Groucho and also a TCF homolog, and they also suggest that these proteins are forming repressive condensates through phase separation.

The authors have produced an extensive dataset to support their two primary claims: that NHR-67 expression levels determine whether a cell is invasive or proliferative, and also that NHR-67 forms a repressive complex through interactions with other proteins. The authors should be commended for clearly and honestly conveying what is already known in this area of study with exhaustive references. Future data unambiguously linking the formation and dissolution of NHR-67 condensates with the activation of downstream genes that NHR-67 is actively repressing would be of great interest to the transcriptional research community.

---

## [Referee Report · Reviewer #2 (Public Review)]

Medwig-Kinney et al. explore the role of the transcription factor NHR-67 in distinguishing between AC and VU cell identity in the *C. elegans* gonad. NHR-67 is expressed at high levels in AC cells where it induces G1 arrest, a requirement for the AC fate invasion program (Matus et al., 2015). NHR-67 is also present at low levels in the non-invasive VU cells and, in this new study, the authors suggest a role for this residual NHR-67 in maintaining VU cell fate. What this new role entails, however, is not clear.

The authors present two models: (1) That NHR-67 switches from a transcriptional activator in ACs to a transcriptional repressor in VUs by virtue of recruiting translational repressors, or (2) that these interactions sequester NHR-67 away from its transcription targets in VU cells. Neither model is fully supported by the data, leaving a paper with extensive data but no single compelling conclusions, and leaving open the question of what is the function, if any, of NHR-67 condensates in VU cells?

While the authors report on interesting observations, in particular the co-localization of NHR-67 with UNC-37/Groucho and POP-1 in nuclear puncta, the functional significance of these observations remains unclear. The authors have not demonstrated that the "repressive condensates" are functional and play a role in the suppression of AC fate in VU cells as claimed. The colocalization data suggest that NHR-67 interacts with repressors, but additional experiments are needed to demonstrate that these interactions are specific to VUs, impact VU fate, and sequester NHR-67 from its targets or transform NHR-67 into a transcriptional repressor.

[Editor's note: we feel that the current state of the data with respect to this question is best captured in the response by the authors to the original concerns expressed by reviewer 2, which we include in abbreviated form here]

1. The authors report that NHR-67 forms "repressive condensates" (aka. puncta) in the nuclei of VU cells and imply that these condensates prevent VU cells from becoming ACs. However, there are also examples of AC cells presented that have NHR-67 puncta (these are less obvious simply due to the higher levels of NHR-67 in ACs). Similarly, there also are UNC-37 and LSY-22 also puncta in ACs. The presence of NHR-67 puncta in the AC seems to directly contradict the author's assumption that the puncta repress the AC fate.

RESPONSE: The puncta formed by NHR-67 in the AC are different in appearance than those observed in the VU cells and furthermore do not exhibit strong colocalization with that of UNC-37 or LSY-22. The Manders' overlap coefficient between NHR-67 and UNC-37 is 0.181 in the AC, whereas it is 0.686 in the VU cells. Likewise, the Manders' overlap coefficient between NHR-67 and LSY-22 is 0.189 in the AC compared to 0.741 in the VU cells. We speculate that the areas of NHR-67 subnuclear enrichment in the AC may represent concentration around transcriptional targets, but testing this would require knowledge of direct targets of NHR-67.

1. While a pool of NHR-67 localizes to "repressive condensates", it appears that a substantial portion of NHR-67 also exists diffusively in the nucleoplasm. This would appear to contradict a "sequestration model" since, for such a model to work, a majority of NHR-67 should be in puncta? What proportion of NHR-67 is in puncta? Is the concentration of NHR-67 in the nucleoplasm lower in VUs compared to ACs and does this depend on the puncta?

RESPONSE: The proportion of NHR-67 localizing to puncta versus the nucleoplasm is dynamic, as these puncta form and dissolve over the course of the cell cycle. However, we estimate that approximately 25-40% of NHR-67 protein resides in puncta based on segmentation and quantification of fluorescent intensity. We also measured NHR-67 concentration in the nucleoplasm of VU cells and found that it is only 28% of what is observed in ACs (n = 10). We also disagree with the notion that the majority of NHR-67 protein should be located in puncta to support the sequestration model. As one example, previously published work examining phase separation of endogenous YAP shows that it is present in the nucleoplasm in addition to puncta (Cai et al., 2019, doi: 10.1038/s41556-019-0433-z). In our system, it is possible that the combination of transcriptional downregulation and partial sequestration away from DNA is sufficient to disrupt the normal activity of NHR-67.

1. The authors do not report whether NHR-67, UNC-37, LSY-22, or POP-1 localization to puncta is interdependent, as implied by their model.

RESPONSE: We based our model, shown in Fig. 7E, on known or predicted protein-protein interactions, which we confirmed through yeast two-hybrid analyses (Fig. 7D; Fig. 7-figure supplement 1). It is difficult to test whether localization of these proteins to puncta is interdependent, as a perturbation of UNC-37, LSY-22, and POP-1 result in ectopic ACs. Trying to determine if loss of puncta results in VU-to-AC transdifferentiation or vice versa becomes a chicken-egg argument. It is also possible that UNC-37 and LSY-22 are at least partially redundant in this context.

1. The evidence that the "repressor condensates" suppress AC fate in VUs is presented in Fig. 4D where the authors deplete the presumed repressor LSY-22. First, the authors do not examine whether NHR-67 forms puncta under these conditions. Second, the authors rely on a single marker (cdh-3p::mCherry::moeABD) to score AC fate: this marker shows weak expression in cells flanking one bright cell (presumably the AC) which the authors interpret as a VU AC transformation. The authors, however, do not identify the cells that express the marker by lineage analyses and dismiss the possibility that the marker-positive cells could arise from the division of an AC-committed cell. Finally, the authors did not test whether marker expression was dependent on NHR-67, as predicted by the model shown in Fig. 7.

RESPONSE: For the auxin-inducible degron experiments, strains contained labeled AID-tagged proteins, a labeled TIR1 transgene, and a labeled AC marker. Thus, we were limited by the number of fluorescent channels we could covisualize and therefore could not also visualize NHR-67 (to assess for puncta formation) or another AC marker (such as LAG-2). We could have generated an AID-tagged LSY-22 strain without a fluorescent protein, but then we would not be able to quantify its depletion, which this reviewer points out is important to measure. We did visualize NHR-67::GFP expression following RNAi-induced knockdown of POP-1 and observed consistent loss of puncta in ectopic ACs. However, it is unclear whether cell fate change causes loss of puncta or vice-versa.

1. Interaction between NHR-67 and UNC-37 is shown using Y2H, but not verified in vivo. Furthermore, the functional significance of the NHR-67/UNC-37 interaction is not tested.

RESPONSE: We attempted to remove the intrinsically disordered region found at the C-terminus of the endogenous nhr-67 locus, using CRISPR/Cas9, as this would both confirm the NHR-67/UNC-37 interaction in vivo and allow us to determine the functional significance of this interaction. However, we were unable to recover a viable line after several attempts, suggesting that this region of the protein is vital.

1. Throughout the manuscript, the authors do not use lineage analysis to confirm fate transformation as is the standard in the field. There are 4 multipotential gonadal cells with the potential to differentiate into VUs or ACs. Which ones contribute to the extra ACs in the different genetic backgrounds examined was not determined, which complicates interpretation. The authors should consider and test the following possibilities: disruption of NHR-67 regulation causes (1) extra pluripotent cells to directly become ACs early in development, (2) causes VU cells to gradually trans-fate to an AC-like fate after VU fate specification (as implied by the authors), or (3) causes an AC to undergo extra cell division(s)? In Fig. 1F, 5 cells are designated as ACs, which is one more that the 4 precursors depicted in Fig. 1A, implying that some of the "ACs" were derived from progenitors that divided.

RESPONSE: The timing between AC/VU cell fate specification and AC invasion (the point at which we look for differentiated ACs) is approximately 10-12 hours at 25 °C. With our imaging setup, we are limited to approximately 3-4 hours of live-cell imaging. Therefore, lineage tracing was not feasible for our experiments. Instead, we relied on visualization of established markers of AC and VU cell fate to determine how ectopic ACs arose. In Fig. 6B,C we show that the expression of two AC markers (cdh-3 and lag-2) turn on while a VU marker (lag-1) gets downregulated within the same cell. In our opinion, live-imaging experiments that show in real time changes in cell fate via reporters was the most definitive way to observe the phenotype.

1. There are 4 multipotential gonadal cells with the potential to differentiate into VUs or ACs. Which ones contribute to the extra ACs in the different genetic backgrounds examined was not determined, which complicates interpretation. The authors should consider and test the following possibilities: disruption of NHR-67 regulation causes (1) extra pluripotent cells to directly become ACs early in development, (2) causes VU cells to gradually trans-fate to an AC-like fate after VU fate specification (as implied by the authors), or (3) causes an AC to undergo extra cell division(s)?? In Fig. 1F, 5 cells are designated as ACs, which is one more that the 4 precursors depicted in Fig. 1A, implying that some of the "ACs" were derived from progenitors that divided.

RESPONSE: When trying to determine the source of the ectopic ACs, we considered the three possibilities noted by the reviewer: (1) misspecification of AC/VU precursors, (2) VU-to-AC transdifferentiation, or (3) proliferation of the AC. We eliminated option 3 as a possibility, as the ectopic ACs we observed here were invasive and all of our previous work has shown that proliferating ACs cannot invade and that cell cycle exit is necessary for invasion (Matus et al., 2015; MedwigKinney & Smith et al., 2020; Smith et al., 2022). Specifically, NHR-67 is upstream of the cyclin dependent kinase CKI-1 and we found that induced expression of NHR-67 resulted in slow growth and developmental arrest, likely because of inducing cell cycle exit. For our experiment using hsp::NHR-67, we induced heat shock after AC/VU specification. For POP-1 perturbation, we explicitly acknowledged that misspecification of the AC/VU precursors could also contribute to ectopic ACs (Fig. 6A; lines 368-385). We could not achieve robust protein depletion through delayed RNAi treatment, so instead we utilized timelapse microscopy and quantification of AC and VU cell markers (Fig. 6B,C; see response 2.7 above).

---

## [Author Response]

**Reviewer #1 (Public Review):**
Medwig-Kinney et al perform the latest in a series of studies unraveling the genetic and physical mechanisms involved in the formation of *C. elegans* gonad. They have paid particular attention to how two different cell fates are specified, the ventral uterine (VU) or anchor cell (AC), and the behaviors of these two cell types. This cell fate choice is interesting because the anchor cell performs an invasive migration through a basement membrane. A process that is required for correct *C. elegans* gonad formation and that can act as a model for other invasive processes, such as malignant cancer progression. The authors have identified a range of genes that are involved in the AC/VC fate choice, and that imparts the AC cell with its ability to arrest the cell cycle and perform an invasive migration. Taking advantage of a range of genetic tools, the authors show that the transcription factor NHR-63 is strongly expressed in the AC cell. The authors also present evidence that NHR-63 is could function as a transcriptional repressor through interactions with a Groucho and also a TCF homolog, and they also suggest that these proteins are forming repressive condensates through phase separation.The authors have produced an extensive dataset to support their two primary claims: that NHR-67 expression levels determine whether a cell is invasive or proliferative, and also that NHR-67 forms a repressive complex through interactions with other proteins. The authors should be commended for clearly and honestly conveying what is already known in this area of study with exhaustive references. But absent data unambiguously linking the formation and dissolution of NHR-67 condensates with the activation of downstream genes that NHR-67 is actively repressing, the novelty of these findings is limited.

Response 1.1: We thank the reviewer for recognizing the extensive dataset we provide in this manuscript in support of our claims that, (1) NHR-67 expression levels are important for distinguishing between AC and VU cell fates, and (2) NHR-67 interacts with transcriptional repressors in VU cells. We acknowledge that a complete mechanistic understanding of the functional significance of NHR-67 puncta is not possible without knowing direct targets of NHR-67 in the AC. Unfortunately, tools to identify transcriptional targets in individual cells or lineages in *C. elegans* do not exist, and generation of such tools would be beyond the scope of this work. This is evidenced by the fact that the first successful attempt to transcriptionally profile the AC was only posted as a preprint one month ago (Costa et al., doi:10.1101/2022.12.28.522136). It is our hope that the findings we present here can be integrated with future AC- and VUspecific profiling efforts to provide a more complete picture of the functional significance of NHR-67 subnuclear organization.

**Reviewer #2 (Public Review):**
Medwig-Kinney et al. explore the role of the transcription factor NHR-67 in distinguishing between AC and VU cell identity in the *C. elegans* gonad. NHR-67 is expressed at high levels in AC cells where it induces G1 arrest, a requirement for the AC fate invasion program (Matus et al., 2015). NHR-67 is also present at low levels in the non-invasive VU cells and, in this new study, the authors suggest a role for this residual NHR-67 in maintaining VU cell fate. What this new role entails, however, is not clear. The model in Figure 7E shows NHR-67 switching from a transcriptional activator in ACs to a transcriptional repressor in VUs by virtue of recruiting translational repressors. In this model, NHR-67 actively suppresses AC differentiation in VU cells by binding to its normal targets and acting as a repressor rather than an activator. Elsewhere in the text, however, the authors suggest that NHR-67 is "post-translationally sequestered" (line 450) in nuclear condensates in VU cells. In that model, the low levels of NHR-67 in VU cells are not functional because inactivated by sequestration in condensates away from DNA. Neither model is fully supported by the data, which may explain why the authors seem to imply both possibilities. This uncertainty is confusing and prevents the paper from arriving at a compelling conclusion. What is the function, if any, of NHR-67 and so-called "repressive condensates" in VU cells?

Response 2.1: As the reviewer correctly notes, we present two possible models in this manuscript. The interaction between NHR-67 and the Groucho/TCF complex in the VU cells could (1) switch the role of NHR-67 from a transcriptional activator to a transcriptional repressor, or (2) sequester NHR-67 away from its transcriptional targets. Indeed, we cannot definitively exclude the possibility of either model. In our resubmission, we will attempt to make this more clear in the text and by presenting both possible models in the summary figure (Fig. 7E).

Below we list problems with data interpretation and key missing experiments:1. The authors report that NHR-67 forms "repressive condensates" (aka. puncta) in the nuclei of VU cells and imply that these condensates prevent VU cells from becoming ACs. Fig. 3A, however, shows an example of an AC that also assemble NHR-67 puncta (these are less obvious simply due to the higher levels of NHR-67 in ACs). The presence of NHR-67 puncta in the AC seems to directly contradict the author's assumption that the puncta repress the AC fate program. Similarly, Figure 5-figure supplement 1A shows that UNC-37 and LSY-22 also form puncta in ACs. The authors need to analyze both AC and VU cells to demonstrate that NHR-67 puncta only form in VUs, as implied by their model.

Response 2.2: The puncta formed by NHR-67 in the AC are different in appearance than those observed in the VU cells and furthermore do not exhibit strong colocalization with that of UNC-37 or LSY-22. The Manders’ overlap coefficient between NHR-67 and UNC-37 is 0.181 in the AC, whereas it is 0.686 in the VU cells. Likewise, the Manders’ overlap coefficient between NHR-67 and LSY-22 is 0.189 in the AC compared to 0.741 in the VU cells. We speculate that the areas of NHR-67 subnuclear enrichment in the AC may represent concentration around transcriptional targets, but testing this would require knowledge of direct targets of NHR-67.

2. While a pool of NHR-67 localizes to "repressive condensates", it appears that a substantial portion of NHR-67 also exists diffusively in the nucleoplasm. This would appear to contradict a "sequestration model" since, for such a model to work, a majority of NHR-67 should be in puncta. What proportion of NHR-67 is in puncta? Is the concentration of NHR-67 in the nucleoplasm lower in VUs compared to ACs and does this depend on the puncta?

Response 2.3: The proportion of NHR-67 localizing to puncta versus the nucleoplasm is dynamic, as these puncta form and dissolve over the course of the cell cycle. However, we estimate that approximately 25-40% of NHR-67 protein resides in puncta based on segmentation and quantification of fluorescent intensity of sum Z-projections. We also measured NHR-67 concentration in the nucleoplasm of VU cells and found that it is only 28% of what is observed in ACs (n = 10). We disagree with the notion that the majority of NHR-67 protein should be located in puncta to support the sequestration model. As one example, previously published work examining phase separation of endogenous YAP shows that it is present in the nucleoplasm in addition to puncta (Cai et al., 2019, doi: 10.1038/s41556-019-0433-z). In our system, it is possible that the combination of transcriptional downregulation and partial sequestration away from DNA is sufficient to disrupt the normal activity of NHR-67.

3. The authors do not report whether NHR-67, UNC-37, LSY-22, or POP-1 localization to puncta is interdependent, as implied in the model shown in Fig. 7.

Response 2.4: It is difficult to test whether localization of these proteins to puncta is interdependent, as perturbation of UNC-37, LSY-22, and POP-1 result in ectopic ACs. Trying to determine if loss of puncta results in VU-to-AC transdifferentiation or vice versa becomes a chicken-egg argument. It is also possible that UNC-37 and LSY-22 are at least partially redundant in this context. We based our model, shown in Fig. 7E, on known or predicted protein-protein interactions, which we confirmed through yeast two-hybrid analyses (Fig. 7D; Fig. 7-figure supplement 1).

4. The evidence that the "repressor condensates" suppress AC fate in VUs is presented in Fig. 4D where the authors deplete the presumed repressor LSY-22. First, the authors do not examine whether NHR-67 forms puncta under these conditions. Second, the authors rely on a single marker (cdh-3p::mCherry::moeABD) to score AC fate: this marker shows weak expression in cells flanking one bright cell (presumably the AC) which the authors interpret as a VU AC transformation. The authors, however, do not identify the cells that express the marker by lineage analyses and dismiss the possibility that the marker-positive cells could arise from the division of an ACcommitted cell. Finally, the authors did not test whether marker expression was dependent on NHR-67, as predicted by the model shown in Fig. 7.

Response 2.5: For the auxin-inducible degron experiments, strains contained labeled AID-tagged proteins, a labeled TIR1 transgene, and a labeled AC marker. Thus, we were limited by the number of fluorescent channels we could covisualize and therefore could not also visualize NHR-67 (to assess for puncta formation) or another AC marker (such as LAG-2). We could have generated an AID-tagged LSY-22 strain without a fluorescent protein, but then we would not be able to quantify its depletion, which this reviewer points out is important to measure. We did visualize NHR-67::GFP expression following RNAi-induced knockdown of POP-1 and observed consistent loss of puncta in ectopic ACs. However, this again becomes a chicken-egg argument as far as whether cell fate change or loss of puncta causes the other.

5. Interaction between NHR-67 and UNC-37 is shown using Y2H, but not verified in vivo. Furthermore, the functional significance of the NHR-67/UNC-37 interaction is not tested.

Response 2.6: We attempted to remove the intrinsically disordered region found at the C-terminus of the endogenous nhr-67 locus, using CRISPR/Cas9, as this would both confirm the NHR-67/UNC-37 interaction in vivo and allow us to determine the functional significance of this interaction. However, we were unable to recover a viable line after several attempts, suggesting that this region of the protein is vital.

6. Throughout the manuscript, the authors do not use lineage analysis to confirm fate transformation as is the standard in the field.

Response 2.7: The timing between AC/VU cell fate specification and AC invasion (the point at which we look for differentiated ACs) is approximately 10-12 hours at 25 °C. With our imaging setup, we are limited to approximately 3-4 hours of live-cell imaging. Therefore, lineage tracing was not feasible for our experiments. Instead, we relied on visualization of established markers of AC and VU cell fate to determine how ectopic ACs arose. In Fig. 6B,C we show that the expression of two AC markers (cdh-3 and lag-2) turn on while a VU marker (lag-1) get downregulated within the same cell. In our opinion, live-imaging experiments that show in real time changes in cell fate via reporters was the most definitive way to observe the phenotype.

There are 4 multipotential gonadal cells with the potential to differentiate into VUs or ACs. Which ones contribute to the extra ACs in the different genetic backgrounds examined was not determined, which complicates interpretation. The authors should consider and test the following possibilities: disruption of NHR-67 regulation causes (1) extra pluripotent cells to directly become ACs early in development, (2) causes VU cells to gradually trans-fate to an AC-like fate after VU fate specification (as implied by the authors), or (3) causes an AC to undergo extra cell division(s)?? In Fig. 1F, 5 cells are designated as ACs, which is one more that the 4 precursors depicted in Fig. 1A, implying that some of the "ACs" were derived from progenitors that divided.

Response 2.8: When trying to determine the source of the ectopic ACs, we considered the three possibilities noted by the reviewer: (1) misspecification of AC/VU precursors, (2) VU-to-AC transdifferentiation, or (3) proliferation of the AC. We eliminated option 3 as a possibility, as the ectopic ACs we observed here were invasive and all of our previous work has shown that proliferating ACs cannot invade and that cell cycle exit is necessary for invasion (Matus et al., 2015; MedwigKinney & Smith et al., 2020; Smith et al., 2022). Specifically, NHR-67 is upstream of the cyclin dependent kinase CKI-1 and we found that induced expression of NHR-67 resulted in slow growth and developmental arrest, likely because of inducing cell cycle exit. For our experiment using hsp::NHR-67, we induced heat shock after AC/VU specification. For POP-1 perturbation, we explicitly acknowledged that misspecification of the AC/VU precursors could also contribute to ectopic ACs (Fig. 6A; lines 368-385). We could not achieve robust protein depletion through delayed RNAi treatment, so instead we utilized timelapse microscopy and quantification of AC and VU cell markers (Fig. 6B,C; see response 2.7 above).

In conclusion, while the authors report on interesting observations, in particular the co-localization of NHR-67 with UNC-37/Groucho and POP-1 in nuclear puncta, the functional significance of these observations remains unclear. The authors have not demonstrated that the "repressive condensates" are functional and play a role in the suppression of AC fate in VU cells as claimed. The colocalization data suggest that NHR-67 interacts with repressors, but additional experiments are needed to demonstrate that these interactions are specific to VUs, impact VU fate, and sequester NHR-67 from its targets or transform NHR-67 into a transcriptional repressor.

Response 2.9: We agree that, at this time, we cannot pinpoint the precise mechanism through which NHR-67 puncta function (i.e., by sequestering NHR-67 from DNA or switching the role of NHR-67 from activating to repressing). However, identification of NHR-67 puncta and their colocalization with UNC-37, LSY-22, and POP-1 in VU cells allowed us to discover an undescribed role for the Groucho/TCF complex in maintaining VU cell fate. This, combined with our evidence demonstrating that NHR-67 transcriptional regulation is important for distinguishing between AC and VU cell fate, are the main contributions of our study.

**Reviewer #1 (Recommendations For The Authors):**
I am not a *C. elegans* researcher and I find this paper fairly hard to follow. One major recommendation I would like to see is to improve the consistency of the labeling of the figures. There are many figures showing many things and I struggled to keep track of everything. For example, the thing that we are looking at in the microscope images (typically GFP tagged to a protein of interest) is sometimes labeled above the image, sometimes to the side, and sometimes within the panel. Experimental conditions are also formatted arbitrarily. As much as they can do so, could the authors try and make their labeling consistent? This would help me follow the data.

Response 1.2: We thank the reviewer for this suggestion and have reorganized the figures (namely Figure 3, Figure 4, Figure 4–figure supplement 1, Figure 5, and Figure 6) such that the tagged allele or marker is labeled at the top, and the time, stage, and/or perturbation is labeled on the side.

Is the yeast one-hybrid assay enough to confirm a direct interaction between HLH-2 and NHR-67? Obviously, it supports it, but since this is not a definitive test in *C. elegans*, I feel the description of this result should be modified to account for this.

Response 1.3: We agree that the yeast one-hybrid assay identifies sequences that are capable of being bound to a protein and does not prove that a DNA-protein interaction occurs in vivo. We have modified our language describing this result in our resubmission (lines 222-224).

NHR-67 and POP-1 eventually form two large spots. This observation supports the claims that these are condensates, but it is clearly different from the observations in Ciona where the condensates remain more or less stable until they quickly dissolve at the onset of mitosis. Do the authors have any idea why these condensates are behaving this way? Is it always two spots? This implies it is forming around some sort of diploid nuclear structure.

Response 1.4: Hes.a puncta observed in Ciona were indeed shown to be dynamic, as puncta were captured fusing together (see Figure 6B of Treen et al., 2021). However, these puncta did not appear to coalesce into two puncta specifically, as is consistently observed with NHR-67 in *C. elegans*. We agree with the reviewer in that this observation is very interesting and likely correlates to a diploid nuclear structure, however we have yet to identify this.

In Ciona, for the two examples of repressive condensates, it was shown that the removal of the C-terminal Groucho recruiting repressor domains of HesA end ERF disrupts condensate formation. Have the authors attempted a similar experiment for NHR-67 or Pop1?

Response 1.5: We agree that this would have been an ideal experiment to perform. We attempted to remove the intrinsically disordered region found at the C-terminus of NHR-67 with CRISPR, but were unable to generate a stable line, suggesting that this region may be critical for NHR-67 function in other developmental stages or tissues.

Other minor points:Fig 4D - I found the labeling of this figure the most confusing.

Response 1.6: We thank the reviewer for bringing this to our attention. For this panel, in addition to the changes we made reference above (Response 1.2), we simplified the labeling of the TIR1 transgene and instead reference it in the figure legend for simplicity.

Line 354 - I think this is mislabeled. Is it supposed to be Figure 5H, not 5F, and 5B, not 5C?

Response 1.7: We thank the reviewer for spotting this error. This reference to Figure 5F has been updated and now correctly references Figure 5H (line 338).

**Reviewer #2 (Recommendations For The Authors):**
The authors use several methods to overexpress NHR-67 including (1) an NHR-67 transgene (Fig. 1), (2) overexpression of the transcriptional activator HLH-2 or (3) removal of a factor that normally degrades HLH-2 in VU cells (Fig. 2). In all cases, the rate of VU AC transformation is either very low (5%) or not reported but presumed to be zero, since other groups have done similar experiments and reported no such conversion (eg. Benavidez et al., 2022). What is the significance of this finding? Does this mean that high levels of NHR-67 are not sufficient to promote AC fate because NHR-67 is sequestered in puncta when expressed in VU cells? Fig. 2A suggests that NHR-67 is in puncta in all VUs where overexpressed. Would the inactivation of GROUCHO in that background result in extra ACs?

Response 2.10: Indeed, we would expect that overexpression of NHR-67 may not normally be sufficient to induce cell fate transformation if the Groucho/TCF complex is still functional. Unfortunately we were unable to achieve strong depletion of UNC-37 and LSY-22 through RNAi, and thus relied on the auxin-inducible protein degradation system. Since we are limited by the number of fluorescent channels we can co-visualize, it would not be feasible to combine a heat-shock inducible transgene, a TIR1 transgene, an AID-tagged protein, and multiple cell fate markers.

The data are often presented as numbers of animals with increased or decreased expression of a particular marker, but no quantification of expression is provided. For example, in Figure 1E, 32/35 animals are reported to exhibit ectopic expression of LIN-12 in the AC and reduced expression of LAG-2. What is the range of the increase/decrease in LIN-12/LAG-2 expression and how does this compare to natural variation in wild-type? The same concerns apply to Fig. 4D.

Response 2.11: For resubmission, we have quantified the data shown in Figure 1E and now report expression levels of LIN-12::mNeonGreen and LAG-2::P2A::H2B::mTurquoise2 in Figure 1–figure supplement 2. We have also quantified the data in Figure 4D and now report expression levels of cdh-3p::mCherry::moeABD in Figure 4E. Quantification methods have been added to the Materials and Methods section (lines 612-617).

The authors explain that it is difficult to study a repressive role for POP-1 as this protein functions in multiple developmental pathways and POP-1 depletion needs to be carefully timed for the data to be interpretable. The authors then go on to use RNAi to deplete POP-1 but do not describe in the methods how they achieve the needed precise temporal control.

Response 2.12: We did indeed describe methods for the GFP-targeting nanobody, which we expressed under a uterinespecific promoter expressed after AC/VU specification. However, since the penetrance of phenotypes associated with this perturbation was low, we utilized RNA interference. We separated the cell fate specification and cell fate maintenance phenotypes by visualizing AC markers (Fig. 6A), which we would expect to be expressed at equal levels if ACs adopted their fate at the same time (via misspecification). We also paired these with a marker for VU cell fate and co-visualized them over time (Fig. 6B,C).

The authors also do not report the efficiency of protein depletion by RNAi or Auxin treatment.

Response 2.13: Auxin-induced depletion of mNeonGreen::AID::LSY-22 resulted in more than 90% decrease in expression (n > 75 uterine cells). The AID-tagged allele for UNC-37 was labeled with BFP, which was barely detectable by our imaging system and photobleached very quickly, so we did not quantify its depletion. However, considering that UNC37 and LSY-22 are both expressed fairly uniform and ubiquitously, and that LSY-22 is expressed at higher levels than UNC-37 at the L3 stage according to WormBase (31.9 FPKM vs. 23.5 FPKM), we would predict that its auxin-induced depletion would be just as potent if not moreso.

Some of the work presented repeats previously published observations, and it is difficult at times to keep track of what is confirmatory and what is new. For example, this group already published on the enrichment of HLH-2 and NHR-67 in the AC, as well as the positive regulation of NHR-67 by HLH-2 (Medwig-Kinney et al 2020). Additionally, prior papers have already reported the interaction between HLH-2 and the nhr-67 locus.

Response 2.14: The work presented in this manuscript does not repeat any previously published experiments. When we introduced the endogenously tagged NHR-67 and HLH-2 strains in previous work (Medwig-Kinney & Smith et al., 2020), we quantified expression of these proteins in the AC over time but did not compare expression between the AC and VU cells. Additionally, we previously showed that HLH-2 positively regulates NHR-67 in the AC (Medwig-Kinney & Smith et al., 2020), but never showed this is the case in the VU cells. Considering that this regulatory interaction was not observed in the AC/VU cell precursors, we believe that determining whether these proteins interact in the context of the VU cells was a valid question to address.

Treen et al. 2021 are cited as prior evidence for the existence of "repressive condensates", however, that study does NOT experimentally demonstrate a function for these structures.

Response 2.15: By “repressive condensates” we are referring to condensation of proteins known to be transcriptional repressors. While we agree that we were not able to demonstrate transcriptional repression of specific loci, our data showing that perturbation of the Groucho repressors UNC-37 and LSY-22 results in ectopic ACs is consistent with the hypothesis that these proteins repress the default AC fate. We have modified our title and text to more clearly distinguish our interpretations versus speculations.